# OMNIEDIT: BUILDING IMAGE EDITING GENERALIST MODELS THROUGH SPECIALIST SUPERVISION

[1,3]**Cong Wei**[*], [2,3]**Zheyang Xiong**[*], [1,3]**Weiming Ren**, [4]**Xinrun Du**, [1,4]**Ge Zhang**, [1,3]**Wenhu Chen**

[1]University of Waterloo, [2]University of Wisconsin-Madison, [3]Vector Institute, [4]M-A-P
cong.wei@uwaterloo.ca, zheyang@cs.wisc.edu, wenhuchen@uwaterloo.ca

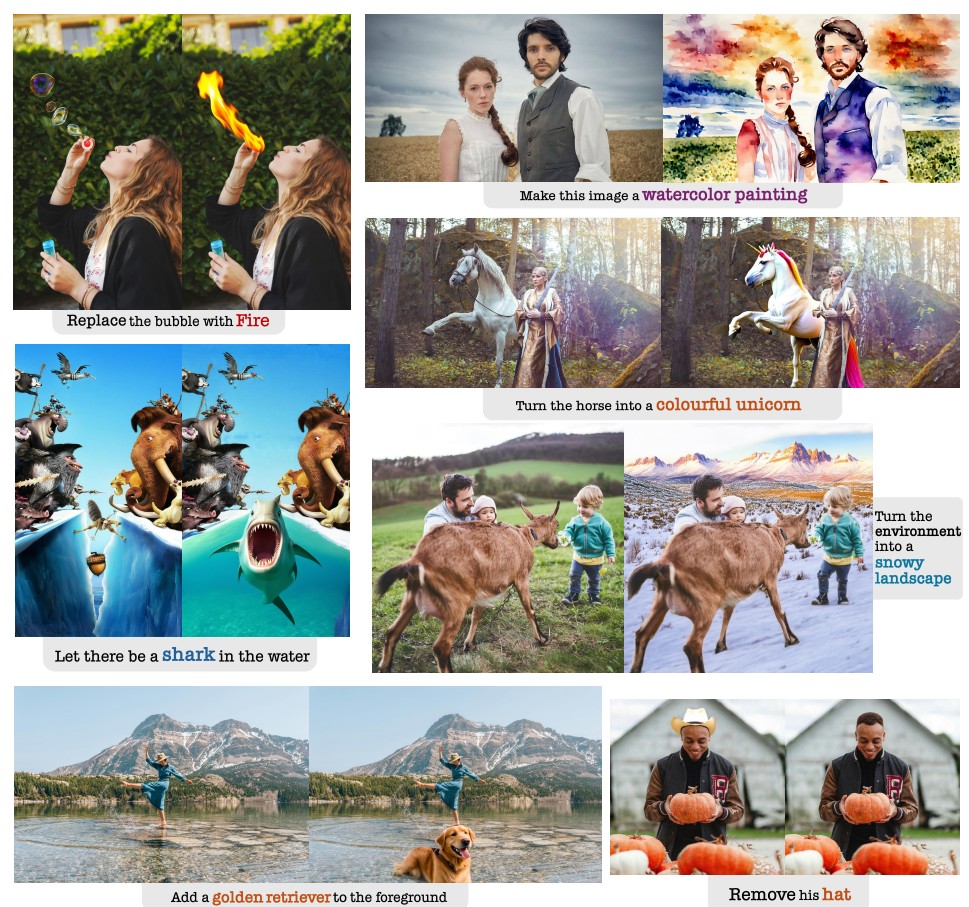

Figure 1: **Editing high-resolution multi-aspect images with OMNI-EDIT.** OMNI-EDIT is an instruction-based image editing generalist capable of performing diverse editing tasks across different aspect ratios and resolutions. It accurately follows instructions while preserving the original image's fidelity. We suggest zooming in for better visualization.

## ABSTRACT

Instruction-guided image editing methods have demonstrated significant potential by training diffusion models on automatically synthesized or manually annotated image editing pairs. However, these methods remain far from practical, real-life applications. We identify three primary challenges contributing to this gap. Firstly, existing models have limited editing skills due to the biased synthesis process. Secondly, these methods are trained with datasets with a high volume of noise and artifacts. This is due to the application of simple filtering methods like CLIP-score. Thirdly, all these datasets are restricted to a single low resolution and fixed aspect ratio, limiting the versatility to handle real-world use cases. In this paper, we present OMNI-EDIT, which is an omnipotent editor to handle

[*]First authors.

seven different image editing tasks with any aspect ratio seamlessly. Our contribution is in four folds: (1) OMNI-EDIT is trained by utilizing the supervision from seven different specialist models to ensure task coverage. (2) we utilize importance sampling based on the scores provided by large multimodal models (like GPT-4o) instead of CLIP-score to improve the data quality. (3) we propose a new editing architecture called EditNet to greatly boost the editing success rate, (4) we provide images with different aspect ratios to ensure that our model can handle any image in the wild. We have curated a test set containing images of different aspect ratios, accompanied by diverse instructions to cover different tasks. Both automatic evaluation and human evaluations demonstrate that OMNI-EDIT can significantly outperform all the existing models. Our code, dataset and model will be available at https://tiger-ai-lab.github.io/OmniEdit/

# 1 INTRODUCTION

Image editing, particularly when following user instructions to apply semantic transformations to real-world photos, has seen significant advancements. Recently, text-guided image editing (Brooks et al., 2023) has gained prominence over traditional methods such as mask-based or region-based editing (Meng et al., 2022). With the rise of diffusion models (Rombach et al., 2022; Podell et al., 2024; Chen et al., 2024a; Sauer et al., 2024), numerous diffusion-based image editing techniques have emerged. Generally, they can be roughly divided into two types: (1) Inversion-based methods (Parmar et al., 2023; Kawar et al., 2023; Gal et al., 2023; Xu et al., 2023; Tumanyan et al., 2023; Tsaban & Passos, 2023) propose to perform zero-shot image editing by inverting the diffusion process and manipulating the attention map in the intermediate diffusion steps to achieve desired editing goal. (2) End-to-end methods (Brooks et al., 2023; Zhang et al., 2024a; Sheynin et al., 2024; Zhao et al., 2024; Fu et al., 2024) propose to fine-tune an existing diffusion model on large-scale image editing pairs to learn the editing operation in an end-to-end fashion. End-to-end methods have generally achieved better performance than inversion-based methods and gained higher popularity.

Table 1: Comparison of OMNI-EDIT with all the existing end-to-end image editing models. The scores are based on a preliminary studies on around 50 prompts.

| Property | InstructP2P | MagicBrush | UltraEdit | MGIE | HQEdit | CosXL | OMNI-EDIT |
|---|---|---|---|---|---|---|---|
| | | | Training Dataset Properties | | | | |
| Real Image? | ✗ | ✓ | ✓ | ✓ | ✗ | ✗ | ✓ |
| Any Res? | ✗ | ✗ | ✗ | ✗ | ✗ | ✗ | ✓ |
| High Res? | ✗ | ✗ | ✗ | ✗ | ✓ | ✗ | ✓ |
| | | | Fine-grained Image Editing Skills | | | | |
| Obj-Swap | ★★☆ | ★★☆ | ★★☆ | ★½☆ | ★★☆ | ★☆☆ | ★★½ |
| Obj-Add | ★☆☆ | ★★☆ | ★☆☆ | ★½☆ | ★☆☆ | ★☆☆ | ★★½ |
| Obj-Remove | ★☆☆ | ★★☆ | ★☆☆ | ★½☆ | ★☆☆ | ★☆☆ | ★★½ |
| Attribute | ★★☆ | ★☆☆ | ★★☆ | ★½☆ | ★☆☆ | ★☆☆ | ★★½ |
| Back-Swap | ★★☆ | ★★☆ | ★★☆ | ★½☆ | ★★☆ | ★★☆ | ★★½ |
| Environment | ★☆☆ | ★☆☆ | ★☆☆ | ★½☆ | ★☆☆ | ★★☆ | ★★½ |
| Style | ★★☆ | ★☆☆ | ★★☆ | ★½☆ | ★☆☆ | ★★½ | ★★½ |

Despite their effectiveness, end-to-end methods face a significant limitation: the scarcity of human-annotated image editing pairs. As a result, all current end-to-end approaches depend on synthetic training data. For instance, existing datasets are synthesized using techniques such as Prompt2Prompt (Hertz et al., 2023) or mask-based editing models like SD-Inpaint (Rombach et al., 2022), and DALLE-2/3 (Ramesh et al., 2022; Betker et al., 2023). However, these synthetic data generation pipelines exhibit significant biases, resulting in the following limitations:

Limited Editing Capabilities: The synthetic data is heavily influenced by the underlying generation models. For example, Prompt2Prompt struggles with localized edits, such as adding, removing, or swapping objects, while SD-Inpaint and DALLE-2 are ineffective at global edits, such as style or background changes. As a result, models trained on such data inherit these limitations.

Poor Data Quality Control: Most approaches use simplified filtering mechanisms like CLIP-score (Radford et al., 2021) or DINO-score (Caron et al., 2021) to automatically select training samples. However, recent studies (Ku et al., 2024) show that these metrics exhibit poor correlation with actual data quality, leading to suboptimal training data that negatively impacts the model.

Lack of Support for Varying Resolutions: All current models are trained on square image editing pairs, making their generalization to non-square images poor.

In our preliminary studies, we curate a few prompts for seven different desired tasks to observe their success rate across the board. We show our findings in Table 1. This show that these models are truly biased in their skills caused by the underlying synthesis pipeline.

In this paper, we introduce OMNI-EDIT, a novel model designed to address these challenges through four key innovations:

**1. Specialist-to-Generalist Supervision:** We propose learning a generalist editing model, OMNI-EDIT, by leveraging supervision from multiple specialist models. Unlike previous approaches that rely on a single expert, we conduct an extensive survey and construct (or train) seven experts, each specializing in a different editing task. These specialists provide supervisory signals to OMNI-EDIT.

**2. Importance Sampling:** To ensure high-quality training data, we employ large multimodal models to assign quality scores to synthesized samples. Given the computational cost of GPT-4o (Achiam et al., 2023), we first distill its scoring ability into InternVL2 (Chen et al., 2024b) through medium-sized samples. Then we use the InternVL2 model for large-scale scoring.

**3. EditNet Architecture:** We introduce EditNet, a novel diffusion-transformer-based architecture (Peebles & Xie, 2022) that facilitates interaction between the control branch and the original branch via intermediate representations. This architecture enhances OMNI-EDIT 's ability to comprehend diverse editing tasks.

**4. Support for Any Aspect Ratio:** During training, we incorporate a mix of images with varying aspect ratios as well as high resolution, ensuring that OMNI-EDIT can handle images of any aspect ratio with any degradation in the output quality.

We curate an image editing benchmark OMNI-EDIT-BENCH, which contains diverse images of different resolutions and diverse prompts that cover all the listed editing skills. We perform comprehensive automatic and human evaluation to show the significant boost of OMNI-EDIT over the existing baseline models like CosXL-Edit (Boesel & Rombach, 2024), UltraEdit (Zhao et al., 2024), etc.

## 2 PRELIMINARIES

### 2.1 TEXT-TO-IMAGE DIFFUSION MODELS

Diffusion models (Song et al., 2021; Ho et al., 2020) are a class of latent variable models parameterized by $\theta$, defined as $p_\theta(\mathbf{x}_0) := \int p_\theta(\mathbf{x}_{0:T}) \, d\mathbf{x}_{1:T}$, where $\mathbf{x}_0 \sim q(\mathbf{x}_0)$ represents the original data, and $\mathbf{x}_1, \ldots, \mathbf{x}_T$ are progressively noisier latent representations of the input image $\mathbf{x}_0$. Throughout the process, the dimensionality of $\mathbf{x}_0$ and the latent variables $\mathbf{x}_{1:T}$ remains consistent, with $\mathbf{x}_{0:T} \in \mathbb{R}^d$, where $d$ corresponds to the product of the image's height, width, and channels. The forward (diffusion) process, denoted as $q(\mathbf{x}_{1:T}|\mathbf{x}_0)$, is a predefined Markov chain that incrementally adds Gaussian noise to the data according to a pre-defined schedule $\{\beta_t\}_{t=1}^T$. The process of forward diffusion is defined as:

$$q(\mathbf{x}_{1:T}|\mathbf{x}_0) = \prod_{t=1}^T q(\mathbf{x}_t|\mathbf{x}_{t-1}), \quad q(\mathbf{x}_t|\mathbf{x}_{t-1}) := \mathcal{N}(\mathbf{x}_t; \sqrt{1-\beta_t}\,\mathbf{x}_{t-1}, \beta_t\mathbf{I}), \tag{1}$$

where $\mathcal{N}$ denotes a Gaussian distribution, and $\beta_t$ controls the amount of noise added at each step. The objective of diffusion models is to reverse this diffusion process by learning the distribution $p_\theta(\mathbf{x}_{t-1}|\mathbf{x}_t)$, which enables the reconstruction of the original data $\mathbf{x}_0$ from a noisy latent $\mathbf{x}_t$. This reduces to a denoising problem where the model $\epsilon_\theta$ is trained to denoise the sample $\mathbf{x}_t \sim q(\mathbf{x}_t|\mathbf{x}_0)$ back into $\mathbf{x}_0$. The maximum log-likelihood training objective breaks down to minimizing the weighted mean squared error between the model's prediction $\hat{\mathbf{x}}_\theta(\mathbf{x}_t, c)$ and the true data $\mathbf{x}_0$:

$$\arg\max_\theta \log p_\theta(\mathbf{x_0}|c) = \arg\min_\theta \mathbb{E}_{(\mathbf{x}_0,c)\sim\mathcal{D}}\left[\mathbb{E}_{\epsilon,t}\left[w_t \cdot \|\hat{\mathbf{x}}_\theta(\mathbf{x}_t, c) - \mathbf{x}_0\|_2^2\right]\right], \tag{2}$$

where $(\mathbf{x}_0, c)$ pairs come from the dataset $\mathcal{D}$, with $c$ representing the text prompt. The term $w_t$ is a weighting factor applied to the loss at each timestep $t$. For simplicity, prior papers (Song et al., 2021; Ho et al., 2020; Karras et al., 2022) will set $w_t$ to be 1.

## 2.2 Instruction-Based Image Editing in Supervised Learning

Instruction-based image editing can be formulated as a supervised learning problem. Existing methods (Brooks et al., 2023; Zhang et al., 2024a) often adopt a paired training dataset of text editing instructions and images before and after the edit. An image editing diffusion model is then trained on this dataset. The latent diffusion objective is defined as:

$$\arg\max_{\theta} \log p_{\theta}(\mathbf{x_0'}|\mathbf{x_0}, c) = \arg\min_{\theta} \mathbb{E}_{(\mathbf{x_0'}, \mathbf{x_0}, c) \sim \mathcal{D}} \left[ \mathbb{E}_{\epsilon, t} \|\hat{\mathbf{x}}_{\theta}(\mathbf{x}_t, c) - \mathbf{x_0'}\|_2^2 \right], \quad (3)$$

where $(\mathbf{x_0'}, \mathbf{x_0}, c)$ triples are sampled from the dataset $\mathcal{D}$ with $\mathbf{x_0}$ denoting the source image, $c$ denoting the editing instruction and $\mathbf{x_0'}$ denoting the target image.

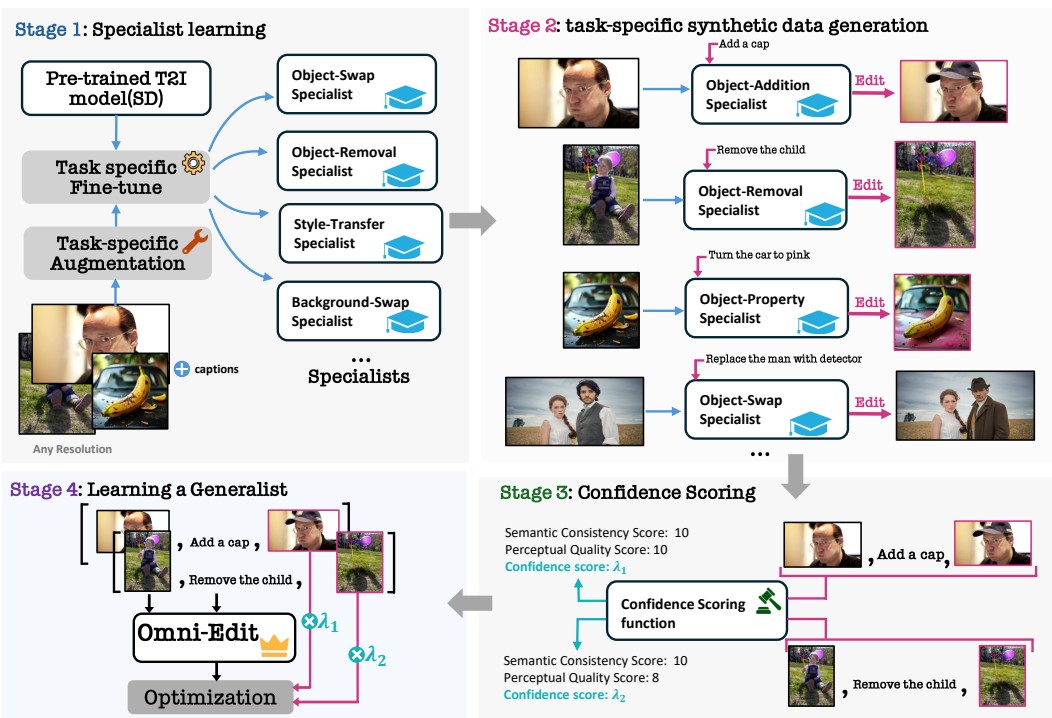

Figure 2: Overview of the OMNI-EDIT training pipeline. The pipeline consists of four stages: (1) task-specific specialist models are trained for diverse editing tasks; (2) these specialist models are used to generate a large, high-resolution, multi-aspect-ratio dataset; (3) a cost-efficient distilled large multi-modal model (LMM) assigns importance weights to each pair of image-editing data; and (4) the final generalist model is trained on the weighted dataset.

## 3 Learning with Specialist Supervision

In this section, we introduce the entire specialist-to-generalist learning framework to build OMNI-EDIT. We describe the overall learning objective in subsection 3.1. We then describe how we learn the specialists in subsection 3.2 and the importance weighting function in subsection 3.3. In Figure 2, we show the overview of the OMNI-EDIT training pipeline.

### 3.1 Learning Objective

We assume there is a groundtruth editing model $p(\mathbf{x'}|\mathbf{x}, c)$, which can perform any type of editing tasks perfectly according to the instruction $c$. Our goal is to minimize the divergence between $p_{\theta}(\mathbf{x'}|\mathbf{x}, c)$ with $p(\mathbf{x'}|\mathbf{x}, c)$ by updating the parameters $\theta$:

$$L(\theta) := \sum_{\mathbf{x}, c} D_{KL}(p(\mathbf{x'}|\mathbf{x}, c)\|p_{\theta}(\mathbf{x'}|\mathbf{x}, c)) = -\sum_{\mathbf{x}, c} \sum_{\mathbf{x'}} p(\mathbf{x'}|\mathbf{x}, c) \log p_{\theta}(\mathbf{x'}|\mathbf{x}, c) + C \quad (4)$$

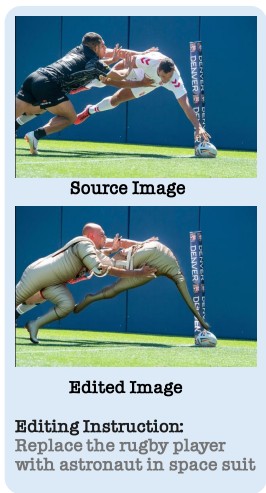

**InternVL2:**
**PQ:**
{ **reasoning**: ….**There are no visible distortions…**The lighting and shadows are consistent,
**score: 10** }
**SC:**
{ **reasoning**: …**The edited image perfectly follows this instruction**. The rugby player is
completely replaced by an astronaut in a space suit... The editing is effective and matches the
instruction without any overediting…,
**score: [10, 10]**}

**InternVL2-Distilled-from-GPT4o:**
**PQ**:
{**reasoning**: The image has some **notable distortions**, **especially in body
proportions**, as the legs and arms of the individuals appear unnaturally elongated…,
**score: 4**}
**SC**:
{**reasoning**: The editing **partially follows the instruction** by adding padded suits that
slightly resemble space suits, **but it doesn't convincingly replace the players with
astronauts**… The players' bodies, poses, and proportions remain largely intact, **making
it feel like an incomplete transformation.**
**score: [5, 6]**}

Source Image

Edited Image

**Editing Instruction:**
Replace the rugby player
with astronaut in space suit

Figure 3: InternVL2 as a scoring function before (top right) and after (bottom right) fine-tuning on
GPT-4o's response. On the top right, the original InternVL2 fails to identify the unusual distortions
in the edited image it also does not spot the error when the edited image fails to meet the specified
editing instructions. On the bottom right, finetuned-InternVL2 successfully detects such failures and
serve as a reliable scoring function.

where $C$ is a constant, which we leave out in the following derivation. However, since we don't
have access to $p(\mathbf{x}'|\mathbf{x}, c)$, we adopt importance sampling for approximation:

$$
\begin{aligned}
L(\theta) &= -\sum_{\mathbf{x},c} \sum_{\mathbf{x}'} q(\mathbf{x}'|\mathbf{x}, c) \frac{p(\mathbf{x}'|\mathbf{x}, c)}{q(\mathbf{x}'|\mathbf{x}, c)} \log p_\theta(\mathbf{x}'|\mathbf{x}, c) \\
&\approx -\mathbb{E}_{(\mathbf{x},c)\sim D} \left[ \mathbb{E}_{\mathbf{x}'\sim q(\mathbf{x}'|\mathbf{x}, c)} \left[ \lambda(\mathbf{x}', \mathbf{x}, c) \log p_\theta(\mathbf{x}'|\mathbf{x}, c) \right] \right] \\
&\approx -\mathbb{E}_{(\mathbf{x},c)\sim D} \left[ \mathbb{E}_{\mathbf{x}'\sim q_s(\mathbf{x}'|\mathbf{x}, c)} \left[ \lambda(\mathbf{x}', \mathbf{x}, c) \log p_\theta(\mathbf{x}'|\mathbf{x}, c) \right] \right]
\end{aligned}
\tag{5}
$$

where $q(\mathbf{x}'|\mathbf{x}, c)$ is the proposal distribution and $\lambda(\cdot)$ is the importance function. To better approx-
imate the groundtruth distribution $p(\mathbf{x}'|\mathbf{x}, c)$, we propose to use an ensemble model $q(\mathbf{x}'|\mathbf{x}, c)$. In
essence, $q(\mathbf{x}'|\mathbf{x}, c) := q_s(\mathbf{x}'|\mathbf{x}, c)$, where $q_s$ is a specialist distribution decided by the type of the
instruction $c$ (e.g. object removal, object addition, stylization, etc). Combing with Equation 3, our
objective can be rewritten as:

$$
\arg\min_\theta L(\theta) = \arg\min_\theta \mathbb{E}_{(\mathbf{x},c)\sim D}\mathbb{E}_{\mathbf{x}'\sim q_s(\mathbf{x}'|\mathbf{x}, c)}\lambda(\mathbf{x}', \mathbf{x}, c) \left[ \mathbb{E}_{\epsilon, t}\|\hat{\mathbf{x}}_\theta(\mathbf{x}_t, \mathbf{x}, c) - \mathbf{x}'\|_2^2 \right].
\tag{6}
$$

The whole process can be described as: we first sample a pair from dataset $D$, and then choose the
corresponding specialist $q_s$ to sample demonstrations $\mathbf{x}'$ for the our editing model $\hat{\mathbf{x}}_\theta(\mathbf{x}_t, \mathbf{x}, c)$ to
approximate with an importance weight of $\lambda(\mathbf{x}', \mathbf{x}, c)$. We formally provide the algorithm in 1. In
our specialist-to-generalist framework, we need to have a series of specialist models $\{q_s(\cdot)\}_s$ and an
importance function $\lambda(\cdot)$. We describe them separately in subsection 3.2 and subsection 3.3.

### 3.2 CONSTRUCTING SPECIALIST MODELS

We group the image editing task into 7 categories as summarized in Table 2. For each category,
we train or build a task specialist $p_s(\mathbf{x}' \mid \mathbf{x}, c)$ to generate millions of examples. Table 2 provides
detailed information on task groups and example editing instructions $c$. In this section, we briefly
summarize each specialist, with details available in Appendix A.1.

**Object Replacement.** We trained an image-inpainting model to serve as the specialist $q_{\text{obj\_replace}}$
for object replacement. Given a image $\mathbf{x}$ and an object caption $c_{\text{obj}}$ and a object mask $M_{\text{obj}}$. The
$q_{\text{obj\_replace}}$ can fill the content indicated by the mask with an object in $c_{\text{obj}}$. We then generate an object
replacement sample by masking out an existing object and fill the image with a new object.
**Object Removal.** We trained an image inpainting model to serve as the specialist $q_{\text{obj\_removal}}$ for
object removal. We use a similar procedure as in the object replacement but use a predicted back-
ground content caption to inpaint the masked image.

Table 2: Task Definitions and Examples

| Editing Tasks | Definition | Instruction $c$ Example |
|---|---|---|
| Object Swap | $c$ describes an object to replace by specifying both the object to remove and the new object to add, along with their properties such as appearance and location. | Replace the black cat with a brown dog in the image. |
| Object Removal | $c$ describes which object to remove by specifying the object's properties such as appearance, location, and size. | Remove the black cat from the image. |
| Object Addition | $c$ describes a new object to add by specifying the object's properties such as appearance and location. | Add a red car to the left side of the image. |
| Attribute Modification | $c$ describes how to modify the properties of an object, such as changing its color and facial expression. | Change the blue car to a red car. |
| Background Swap | $c$ describes how to replace the background of the image, specifying what the new background should be. | Replace the background with a space-ship interior. |
| Environment Change | $c$ describes a change to the overall environment, such as the weather, lighting, or season, without altering specific objects. | Change the scene from daytime to nighttime. |
| Style Transfer | $c$ describes how to apply a specific artistic style or visual effect to the image, altering its overall appearance while keeping the content the same. | Apply a watercolor painting style to the image. |

**Object Addition.** We treat object addition as the inverse task of object removal.

**Attribute Modification.** We adopt the Prompt-to-Prompt (P2P) (Hertz et al., 2023) pipeline to generate examples. To enable precise modification, we adapt the method from Sheynin et al. (2024) where we provide a mask $M_{obj}$ for the object and force P2P to only make edits inside the mask.

**Background Swap.** We use a similar procedure as in the object replacement but use an inverse mask of the object to indicate the background and guide the inpainting.

**Environment Modification.** For environment modification, we use P2P pipeline to generate original and edited image.

**Style Transfer.** We use CosXL-Edit (Boesel & Rombach, 2024) as the specialist model as its training data contains a large number of style transfering examples. We provide CosXL-Edit with $(\mathbf{x}, c)$, and let it generates the edited image $\mathbf{x}'$.

### 3.3 IMPORTANCE WEIGHTING

The importance weighting function $\lambda$ takes as input a tuple of source image, edited image, and editing prompt. Its purpose is to assign higher weights to data points that are more likely to be sampled from the ground truth distribution, and lower weights to the unlikely ones. This is essentially a quality measure to up-weight high-quality samples. Unlike previous work, we do not use CLIP score because prior work (Jiang et al., 2024) has shown its low correlation with human judges. Instead, we propose to use large multimodal models (LMMs) to approximate the weighting function, as they demonstrate strong image understanding. Following VIEScore (Ku et al., 2024), we designed a prompting template for GPT-4o (Achiam et al., 2023) to evaluate the image editing pairs and output a score on a scale from 0 to 10. We then filter out data with a score greater than or equal to 9, so the LMM essentially serves as a binary weighting function:

$$\lambda(\mathbf{x}', \mathbf{x}, c) = \begin{cases} 1, & \text{if LMM}(\text{prompt}, \mathbf{x}', \mathbf{x}, c) \geq 9 \\ 0, & \text{otherwise} \end{cases}$$

Details of the prompt template are provided in the Appendix.

While the GPT-4o is an effective choice for this task, scoring large-scale datasets with millions of examples is extremely costly and time-consuming. Therefore, we employ knowledge distillation from GPT-4o to a smaller 8B model, InternVL2 (Chen et al., 2024b). For each task, we sample 50K data points and instruct GPT-4o to output both a score and a score rationale. We fine-tune InternVL2 on these GPT-4o-generated examples. After fine-tuning, InternVL2 performs as an ideal scoring function due to its smaller size and efficiency. A comparison of the model's performance before and after fine-tuning is presented in the Appendix. Finally, we apply the fine-tuned InternVL2 model to filter data across a dataset with millions of samples. Only examples with a score of $\geq 9$ are retained, resulting in a curated training dataset of 1.2M examples. We visualize InternVL2's response as a scoring function before and after fine-tuning in Figure 3. We observe that fine-tuning InternVL2 on GPT-4o's response effectively turns InternVL2 into a realiable scoring function and it can identify unusual distortions or unsuccessful edit that does not follow the editing instruction. Additional dataset statistics are detailed in the Appendix.

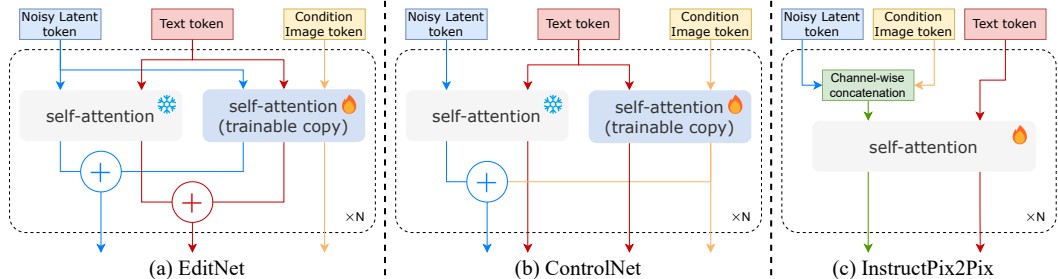

Figure 4: Architecture Comparison between **EditNet(ours)**, ControlNet and InstructPix2Pix(Channel-wise concatenation) for DiT models. Unlike ControlNet's parallel execution, EditNet allows adaptive adjustment of control signals by intermediate representations interaction between the control branch and the original branch. EditNet also updates the text representation, enabling better task understanding.

## 4 EDITNET

We found that directly fine-tuning a pre-trained high-quality diffusion model like SD3 using channel-wise image concatenation methods (Brooks et al., 2023) compromises the model's original representational capabilities (see Figure 10 and Section 5.2 for details comparison).

To enable a diffusion transformer to perform instruction-based image editing while preserving its original capabilities, we introduce **EditNet** to build OMNI-EDIT. EditNet can effectively transform common DIT models like SD3 into editing models. As illustrated in Figure 4, we replicate each layer of the original DIT block as a control branch. The control branch DIT blocks allow interaction between the original DIT tokens, conditional image tokens, and the editing prompts. The output of the control branch tokens is then added to the original DIT tokens and editing prompts. Since the original DIT blocks are trained for generation tasks and are not aware of the editing instructions specifying which contents to modify and how to modify them, this design allows the control branch DIT to adjust the representations of the original DIT tokens and editing prompts according to the editing instruction, while still leveraging the strong generation ability of the original DIT. Compared to ControlNet (Zhang et al., 2023), our approach offers two key advantages that make it more suitable for image editing tasks: First, ControlNet does not update text representations, making it challenging to execute editing tasks based on instruction, particularly object removal, as it fails to understand the "removal" intent (see Figure 11). Secondly, ControlNet's control branch operates in parallel without access to the original branch's intermediate representations. This fixed precomputation of control signals restricts the overall representation power of the network. We provide an ablation study on the OMNI-EDIT architecture design in Section 5.2.

## 5 EXPERIMENTS

In this section, we first provide statistics of the OMNI-EDIT training set and test set in Table 5. Then we introduce the human evaluation protocol in Section 5, and comparative baseline system in 5. We present the main results in Section 5.1, highlighting the advantages of OMNI-EDIT in tacking multi-aspect ratio, multi-resolution, and multi-task image editing. In Section 5.2, we study the advantages of importance sampling for synthetic data. In Section 5.2, we perform an analysis to study the design of OMNI-EDIT.

**OMNI-EDIT Training Dataset.** We constructed the training dataset $\mathcal{D}$ by sampling high-resolution images with a minimum resolution of 1 megapixel from the LAION-5B (Schuhmann et al., 2022) and OpenImageV6 (Kuznetsova et al., 2020) databases. The images cover a range of aspect ratios including 1:1, 2:3, 3:2, 3:4, 4:3, 9:16, and 16:9. For the task of object swap, we employed a specialist model to generate 1.5 million entries. We then applied InternVL2 for importance weighting, retaining samples with scores of 9 or higher, resulting in a dataset of 150K entries for this task. Similarly, we generate 250k-1M samples for each task, then keep the top 10% as the final dataset. The final training dataset comprises 1.2M entries, with detailed information provided in Appendix 4.

**OMNI-EDIT-Bench.** To create a high-resolution, multi-aspect ratio, multi-task benchmark for instruction-based image editing, we manually collected 62 images from pexels (2024) and LAION-

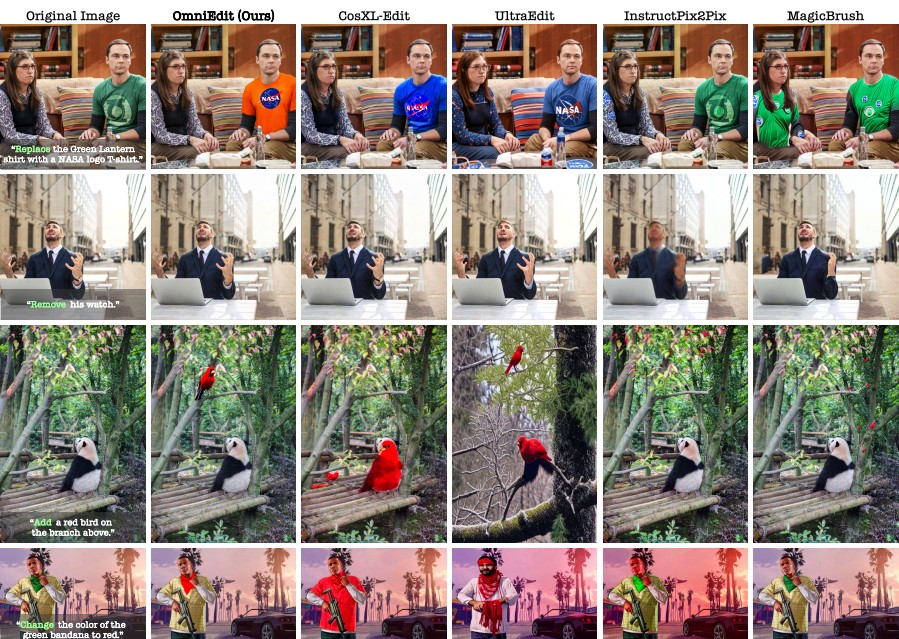

Figure 5: Qualitative comparison between baselines and OMNI-EDIT on a subset of the test set.

5B (Schuhmann et al., 2022). These images cover a variety of aspect ratios, including 1:1, 2:3, 3:2, 3:4, 4:3, 9:16, and 16:9. We ensured that the images feature a diverse range of scenes and object counts, from single to complex compositions. Additionally, we selected images with a relatively high aesthetic score to better align with the practical use cases of image editing. For each image, we tasked the model with performing 7 tasks as outlined in Table 2. This results in a total of 434 edits.

**OMNI-EDIT implementation details.** The OMNI-EDIT model is built upon Stable diffusion 3 Medium(Esser et al., 2024) with EditNet architecture. The stable diffusion 3 has 24 DiT layers. Each layer has a corresponding EditNet layer. We train OMNI-EDIT on the 1.2M OMNI-EDIT training dataset for 2 epochs on a single node with 8 H100 GPUs.

**Baseline models.** We compare OMNI-EDIT with 8 other text-guided image editing baselines: MagicBrush (Zhang et al., 2024a), InstructPix2Pix (Brooks et al., 2023), UltraEdit(SD3) (Zhao et al., 2024), DiffEdit (Couairon et al., 2022), SDEdit (Meng et al., 2022), CosXL-Edit (Boesel & Rombach, 2024), HIVE (Zhang et al., 2024b) and HQ-Edit (Hui et al., 2024).

**Evaluations Protocol** We conduct both human evaluation and automatic evaluation. For the human evaluation, we follow the procedure from Ku et al. (2023) to rate in two criteria: Semantic Consistency ($SC$) and Perceptual Quality ($PQ$). Both scores are in $\{0, 0.5, 1\}$. For $SC$, the human subject is asked to rate the consistency between 1) the edited image and the editing instruction (whether the editing instruction is reflected on the edited image) and between 2) the source image and the edited image (whether the model makes the edit that is beyond the editing instruction). For $PQ$, the subject is asked to rate on the quality of edited image). We then calculate a overall score $O = \sqrt{SC \times PQ}$ that measures the overall quality of the edit. We also calculate the accuracy of the edit, which is defined by the percentage of $SC = 1$ among all examples. We recruit four human raters and require them to evaluate all the editing examples. For LMMs' evaluation, we follow the procedure from Ku et al. (2024) where models (in particular, we chose GPT4o and Gemini) are also asked to give $SC$ and $PQ$ scores but on a scale of 0-10. We then normalize the scale to 0-1.

## 5.1 MAIN RESULTS

We provide a qualitative comparison with baseline models in Figure 5. We show the top 4 baselines with OMNI-EDIT on a subset of the OMNI-EDIT-Bench. We provide more results in Figure 8 and Figure 9. Our main results are detailed in Table 3, where we provide the VIEScore and conduct human evaluation on the Top2 baselines and OMNI-EDIT. In Figure 1, OMNI-EDIT demonstrates its capability to handle diverse editing tasks across various aspect ratios and resolutions. The results are notably sharp and clear, especially in the addition/swap task, where new content is seamlessly integrated. This underscores the effectiveness of the Edit-Net design in preserving the original image

Table 3: Main evaluation results on Omni-Edit-Bench. In each column, the highest score is bolded, and the second-highest is underlined.

| Models | VIEScore (GPT4o) | | | VIEScore (Gemini) | | | Human Evaluation | | | |
|---|---|---|---|---|---|---|---|---|---|---|
| | $PQ_{avg}\uparrow$ | $SC_{avg}\uparrow$ | $O_{avg}\uparrow$ | $PQ_{avg}\uparrow$ | $SC_{avg}\uparrow$ | $O_{avg}\uparrow$ | $PQ_{avg}\uparrow$ | $SC_{avg}\uparrow$ | $O_{avg}\uparrow$ | $Acc_{avg}\uparrow$ |
| Inversion-based Methods | | | | | | | | | | |
| DiffEdit | 5.88 | 2.73 | 2.79 | 6.09 | 2.01 | 2.39 | - | - | - | - |
| SDEdit | 6.71 | 2.18 | 2.78 | 6.31 | 2.06 | 2.48 | - | - | - | - |
| End-to-End Methods | | | | | | | | | | |
| InstructPix2Pix | 7.05 | 3.04 | 3.45 | 6.46 | 1.88 | 2.31 | - | - | - | - |
| MagicBrush | 6.11 | 3.53 | 3.60 | 6.36 | 2.27 | 2.61 | - | - | - | - |
| UltraEdit(SD-3) | 6.44 | 4.66 | 4.86 | 6.49 | 4.33 | 4.45 | 0.72 | 0.52 | 0.57 | 0.20 |
| HQ-Edit | 5.42 | 2.15 | 2.25 | 6.18 | 1.71 | 1.96 | 0.80 | 0.27 | 0.29 | 0.10 |
| CosXL-Edit | 8.34 | 5.81 | 6.00 | 7.01 | 4.90 | 4.81 | 0.82 | 0.56 | 0.59 | 0.35 |
| HIVE | 5.35 | 3.65 | 3.57 | 5.84 | 2.84 | 3.05 | - | - | - | - |
| OMNI-EDIT | **8.38** | **6.66** | **6.98** | **7.06** | **5.82** | **5.78** | **0.83** | **0.71** | **0.69** | **0.55** |
| $\Delta$ - Best baseline | +0.04 | +0.85 | +0.98 | +0.05 | +0.92 | +0.97 | +0.01 | +0.15 | +0.10 | +0.20 |

generation capabilities of the base text-image generative model. Similarly, in Figure 5, OMNI-EDIT uniquely adds a clean and distinct NASA logo onto a T-shirt. Table 3 corroborates this with OMNI-EDIT achieving the highest Perceptual Quality (PQ) score among the models evaluated.

We highlight the efficacy of our proposed specialist-to-generalist learning framework. Unlike baseline models that utilize a single method for generating synthetic data—often the prompt-to-prompt method—This method typically alters the entire image, obscuring task-specific data. In contrast, OMNI-EDIT leverages task-specific data curated by experts, resulting in a clearer task distribution and improved adherence to editing instructions. Both the VIEScore and human evaluations in Table 3 demonstrate that our method significantly outperforms the best baseline in following editing instructions accurately and minimizing over-editing. For instance, baseline models frequently misunderstand the task intent as illustrated in Figure 5, where the CosXL-Edit model fails to recognize the removal task and incorrectly interprets a bird addition as a swap between a panda and a bird.

Lastly, baseline models often produce blurry images on the OMNI-EDIT-Bench, as they are trained at resolutions limited to 512x512 or even 256x256, and they perform poorly on non-square aspect ratios. For example, with a 3:4 aspect ratio, the baselines struggle to perform editing. OMNI-EDIT, trained on data with multiple aspect ratios, maintains robust editing capabilities across the diverse aspect ratios encountered on the Omni-Bench, as evidenced in Figure 5.

## 5.2 ABLATION STUDY

In this section, We provide an ablation study w.r.t importance weighting and EditNet.

**Ablation study on the importance sampling.** We study a baseline that utilizes the same architecture as OMNI-EDIT, but instead of applying importance scoring and filtering, we sample 1.2M examples directly from the 5M pre-filtering dataset as specified in Table 4 and compare it with OMNI-EDIT. As shown in Table 6, we observe a significant decrease in VIEScores for both PQ and SC metrics.

**Ablation Study on OMNI-EDIT Architecture Design.** We conducted an analysis of OMNI-EDIT's architectural design in comparison to two baseline models: OMNI-EDIT-ControlNet and OMNI-EDIT-ControlNet-TextControl and show the result in Table 7. OMNI-EDIT-ControlNet represents the SD3-ControlNet architecture trained on the OMNI-EDIT dataset, where the source image serves as the conditioning image for the control branch. OMNI-EDIT-ControlNet-TextControl is a variant of OMNI-EDIT-ControlNet with an added modification: at each layer, we incorporate the text-token output from the control branch into the text-token in the main image generation branch. So this baseline can update the text representation in the main branch but doesn't have the intermediate representation interaction design in EditNet.

Our analysis, as shown in Figure 11, reveals that OMNI-EDIT-ControlNet struggled to accurately capture task intent. This is primarily because the ControlNet branch does not update the text representation. For instance, in object removal tasks, prompts like "Remove ObjA" are common, yet the original DIT block remains unchanged, causing it to mistakenly generate an image of "ObjA." On the other hand, although OMNI-EDIT-ControlNet-TextControl successfully updates the text representation, it still encounters difficulties in content removal. The substantial VIEScores gap between OMNI-EDIT-Controlnet-TextControl and OMNI-EDIT in Table 7 underscores the importance of the

intermediate representation interaction design in EditNet. We also compared OMNI-EDIT with the channel-wise token concatenation method used in InstructPix2Pix (see Figure 4). Channel-wise Token concatenation requires fine-tuning the entire network, which can distort the network's original representations. As illustrated in Figure 10, after fine-tuning an SD3 channel-wise concatenation model on OMNI-EDIT training set, the representation of Batman is altered. In contrast, EditNet preserves the original representation of Batman while still learning the object swap task.

## 6 RELATED WORK

**Image Editing via Generation.** Editing real images according to specific user requirements has been a longstanding research challenge (Crowson et al., 2022; Liu et al., 2020; Zhang et al., 2023; Shi et al., 2022; Ling et al., 2021; Wasserman et al., 2024; Ju et al., 2024). Since the introduction of large-scale diffusion models, such as Stable Diffusion (Rombach et al., 2022; Podell et al., 2024), significant progress has been made in tackling image editing tasks. SDEdit (Meng et al., 2022) introduced an approach that adds noise to the input image at an intermediate diffusion step, followed by denoising guided by the target text description to generate the edited image. Subsequent methods, such as Prompt-to-Prompt (Hertz et al., 2023) and Null-Text Inversion (Mokady et al., 2023), have focused on manipulating attention maps during intermediate diffusion steps for image editing. Other techniques like Blended Diffusion (Avrahami et al., 2022) and DiffEdit (Couairon et al., 2022) utilize masks to blend regions of the original image into the edited output. More recently, the field has seen a shift towards supervised methods, such as InstructP2P (Brooks et al., 2023), HIVE (Zhang et al., 2024b), and MagicBrush (Zhang et al., 2024a), which incorporate user-written instructions in an end-to-end framework. Our work follows this direction to develop end-to-end editing models without inversion.

**Image Editing Datasets.** Due to the difficulty of collecting expert-annotated editing pairs, existing approaches rely heavily on synthetic data to train editing models. InstructP2P (Brooks et al., 2023) was the first to curate large-scale editing datasets using prompt-to-prompt filtering with CLIP scores. MagicBrush (Zhang et al., 2024a) subsequently improved data quality by incorporating a human-in-the-loop annotation pipeline based on DALLE-2. However, DALLE-2, primarily an inpainting-based method, struggles with global editing tasks such as style transfer and attribute modification. More recently, HQ-Edit (Hui et al., 2024) utilized DALLE-3 to curate editing pairs, although the source and target images lack pixel-to-pixel alignment, which is critical for preserving fine-grained details. Emu Edit (Sheynin et al., 2024) scaled up the training dataset to 10 million proprietary pairs, resulting in strong performance, but the lack of public access to their model checkpoints or API makes direct comparison difficult. UltraEdit (Zhao et al., 2024) proposed another inpainting-based approach, avoiding the use of DALLE-2 or DALLE-3 for data curation. However, like MagicBrush, it still faces limitations in handling complex global edits. Our work is the first to leverage multiple specialists to significantly expand the range of editing capabilities. Additionally, we are the first to use more reliable large multimodal models, for quality control in the editing process.

## 7 DISCUSSION

In this paper, we identify the imbalanced skills in the existing end-to-end image editing methods and propose a new framework to build more omnipotent image editing models. We surveyed the field and chose several approaches as our specialists to synthesize candidate pairs and adopt weighted loss to supervise the single generalist model. Our approach has shown significant quality boost across the broad editing skills. Throughout the experiments, we found that the output quality is highly influenced by the underlying base model. Due to the weakness of SD3, our approach is still not achieving its highest potential. In the future, we plan to use Flux or other more capable base models to see how much further we can reach with the current framework.

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
