## A APPENDIX

Table 4: Omni-Edit training dataset statistics reflecting the number of samples before and after importance scoring and filtering with o-score $\geq 9$.

| Task | Pre-Filtering Number | After-Filtering Number |
|---|---|---|
| Object Swap | 1,500,000 | 180,000 |
| Object Removal | 1,000,000 | 200,000 |
| Object Addition | 1,000,000 | 200,000 |
| Background Swap | 500,000 | 180,000 |
| Environment Change | 500,000 | 160,000 |
| Style Transfer | 250,000 | 50,000 |
| Object Property Modification | 450,000 | 250,000 |
| **Total** | **5,200,000** | **1,220,000** |

---

**Algorithm 1** Specialist-to-Generalist Learning Framework

---

**Require:** Dataset $\mathcal{D} = \{(\mathbf{x_i}, c_i)\}_{i=1}^{N}$ of image-text instruction pairs
**Require:** $\mathcal{K}$ task specialist model $q_k$
**Ensure:** Generalist diffusion model parameterized by $\theta$
 1: **Initialize** a buffer $\mathcal{G} \leftarrow \emptyset$
 2: **for** each pair of $\{(\mathbf{x_s}, c_s)\}$ in $\mathcal{D}$ **do**
 3: $\quad q_s = f(c_s)$, where $f : \mathcal{C} \rightarrow \mathcal{S}$ maps from the instruction space to the set of specialists.
 4: $\quad \mathbf{x'_s} \sim q_s(\mathbf{x'_s}|\mathbf{x_s}, c_s)$.
 5: $\quad$ Compute importance weight $\lambda(\mathbf{x'_s}, \mathbf{x_s}, c_s)$
 6: $\quad \mathcal{G} \leftarrow \mathcal{G} \cup \{(\mathbf{x'_s}, \mathbf{x_s}, c_s), \lambda(\mathbf{x'_s}, \mathbf{x_s}, c_s)\}$
 7: **end for**
 8: Train generalist model $\theta$ on dataset $\mathcal{G}$ using Eq. 6

---

### A.1 TRAINING DATA GENERATION DETAILS

#### A.1.1 OBJECT REPLACEMENT

We trained an image-inpainting model to serve as the expert for object replacement. During training, given a source image $\mathbf{x}_{\text{src}}$ and an object caption $C_{\text{obj}}$, we employ GroundingDINO and SAM to generate an object mask $M_{\text{obj}}$. The masked image is then created by removing the object from the source image:

$$\mathbf{x}_{\text{masked}} = \mathbf{x}_{\text{src}} \odot (1 - M_{\text{obj}}). \tag{7}$$

Here, $\odot$ denotes element-wise multiplication, effectively masking out the object in $\mathbf{x}_{\text{src}}$. Both the mask $M_{\text{obj}}$ and the object caption $C_{\text{obj}}$ are provided as inputs to the expert model $q_{\text{obj\_replace}}$. The expert $q_{\text{obj\_replace}}$ is trained to reconstruct (inpaint) the original source image $\mathbf{x}_{\text{src}}$ from the masked image.

To generate data for object replacement, we sample 200K images from the LAION and OpenImages datasets, ensuring a diverse range of resolutions close to 1 megapixel. For each image, we utilize GPT-4o to propose five object replacement scenarios. Specifically, GPT-4o identifies five interesting source objects $C_{\text{src\_obj}}$ within the image and suggests corresponding target objects $C_{\text{trg\_obj}}$ for replacement.

For each proposed replacement, we perform the following steps:

1. **Mask Generation:** Use GroundingDINO and SAM to generate the object mask $M_{\text{src\_obj}}$ for the source object $C_{\text{src\_obj}}$.

2. **Mask Dilation:** Apply a dilation operation to $M_{\text{src\_obj}}$ to expand the mask boundaries.

3. **Image Editing:** Apply the expert model to generate the edited image $\mathbf{x}_{\text{edit}}$ by replacing the source object with the target object $C_{\text{trg\_obj}}$:

$$\mathbf{x}_{\text{edit}} = q_{\text{obj\_replace}} \left( \mathbf{x}_{\text{src}} \odot (1 - M_{\text{src\_obj}}), \ M_{\text{src\_obj}}, \ C_{\text{trg\_obj}} \right) \tag{8}$$

In this equation:

- $\mathbf{x}_{\text{src}} \odot (1 - M_{\text{src\_obj}})$ represents the source image with the target object masked out.
- $M_{\text{src\_obj}}$ is the mask of the source object to be replaced.
- $C_{\text{trg\_obj}}$ is the caption of the target object for replacement.

Then a pair of instruction-based image editing examples will be: $\langle \mathbf{x}_{\text{src}}, \mathbf{x}_{\text{edit}}, T \rangle$. The instruction $T$ initially just be "Replace $C_{\text{src\_obj}}$ with $C_{\text{trg\_obj}}$ ". We then employ large multimodal models (*LVLM*) to generate more detailed natural language instructions.

### A.1.2 OBJECT REMOVAL

Similar to object replacement, we trained an image inpainting model to serve as the expert for object removal. During training, given a source image $\mathbf{x}_{\text{src}}$ and an image caption $C_{\text{src}}$, we randomly apply strikes to create a mask $M_{\text{src}}$. The masked image is then created by:

$$\mathbf{x}_{\text{masked}} = \mathbf{x}_{\text{src}} \odot (1 - M_{\text{src}}) \tag{9}$$

Both the mask $M_{\text{src}}$ and the image caption $C_{\text{src}}$ are provided as inputs to the expert model $q_{\text{obj\_removal}}$. The expert $q_{\text{obj\_removal}}$ is trained to reconstruct (inpaint) the original source image $\mathbf{x}_{\text{src}}$ from the masked image. To generate data for object removal, we also sample 200K images from the LAION and OpenImages datasets, ensuring a diverse range of resolutions close to 1 megapixel. For each image, we utilize GPT-4o to propose five objects to remove and predict the content of the space after removal. Specifically, GPT-4o identifies five interesting source objects $C_{\text{src\_obj}}$ within the image and predicts the new content after removing the object $C_{\text{trg\_background}}$. For each proposed removal, we perform the following steps:

1. **Mask Generation:** Use GroundingDINO and SAM to generate the object mask $M_{\text{src\_obj}}$ for the source object $C_{\text{src\_obj}}$.

2. **Image Editing:** Apply the expert model to generate the edited image $\mathbf{x}_{\text{edit}}$ by infilling the masked region with the predicted background content $C_{\text{trg\_background}}$:

$$\mathbf{x}_{\text{edit}} = q_{\text{obj\_removal}} \left( \mathbf{x}_{\text{src}} \odot (1 - M_{\text{src\_obj}}), \ M_{\text{src\_obj}}, \ C_{\text{trg\_background}} \right). \tag{10}$$

In this equation:

- $\mathbf{x}_{\text{src}} \odot (1 - M_{\text{src\_obj}})$ represents the source image with the target object masked out.
- $M_{\text{src\_obj}}$ is the mask of the source object to be removed.
- $C_{\text{trg\_background}}$ is the predicted content for the background after object removal.

Then a pair of instruction-based image editing example will be: $\langle \mathbf{x}_{\text{src}}, \mathbf{x}_{\text{edit}}, T \rangle$. Initially, the instruction $T$ initially just be "Remove $C_{\text{src\_obj}}$ from the image" We then employ large multimodal models (*LVLM*) to generate more detailed natural language instructions.

### A.1.3 OBJECT ADDITION

We conceptualize the object addition task as the inverse of the object removal process. Specifically, for each pair of editing examples generated by the object removal expert, we swap the roles of the source and target images to create a new pair tailored for object addition. This approach leverages the naturalness and artifact-free quality of the original source images, ensuring high-quality additions. Given a pair of editing examples $\langle \mathbf{x}_{\text{src\_removal}}, \mathbf{x}_{\text{edit\_removal}}, c_{\text{removal}} \rangle$ generated for object removal and

$C_{\text{src\_obj\_removal}}$ represents the object to remove. We transform this pair into an object addition example by swapping $\mathbf{x}_{\text{src}}$ and $\mathbf{x}_{\text{edit}}$, and modifying the instruction accordingly. The resulting pair for object addition is $\langle \mathbf{x}_{\text{src}} = \mathbf{x}_{\text{edit\_removal}}, \mathbf{x}_{\text{edit}} = \mathbf{x}_{\text{src\_removal}}, c \rangle$, where $c$ is the new instruction defined as "Add $C_{\text{src\_obj\_removal}}$ to the image."

### A.1.4   Attribute Modification

We adapt the Prompt-to-Prompt (P2P) (Hertz et al., 2023) pipeline where a text-guided image generation model is provided with a pair of captions $\langle C_{\text{src}}, C_{\text{edit}} \rangle$ and injects cross-attention maps from the input image generation to that during edited image generation. For example, a pair could be $\langle$ "a blue backpack", "a purple backpack"$\rangle$ with the corresponding editing instruction "make the backpack purple".

To enable precise attribute modification on the object we want (in our example, the "backpack"), we adapt the method from Sheynin et al. (2024) where we provide an additional mask $M_{obj}$ that masks the object. Specifically, to obtain a pair of captions, we obtain source captions $C_{\text{src}}$ from Zhang (2024) and let GPT4 to identify an object $C_{\text{obj}}$ in the original caption $C_{\text{src}}$, propose an editing instruction that edits an attribution of $C_{\text{obj}}$ and output the edited caption $C_{\text{edit}}$ with object's attribution reflected.

We first let the image generation model to generate a source image $\mathbf{x}_{\text{src}}$ using $C_{\text{src}}$. We then use GroundingDINO to extract mask $M_{\text{obj}}$ that masks the object from the source image. We then apply P2P generation with caption pair $\langle C_{\text{src}}, C_{\text{edit}} \rangle$. During the generation, we use the mask to control precise image editing control. In particular, let $\mathbf{x}_{\text{src},t}$ denote the noisy source image at step $t$ and $\mathbf{x}_{\text{edit},t}$ denote the noisy edited image at step $t$, we apply the mask and force the new noisy edited image at time $t$ be $M_{\text{obj}} \odot \mathbf{x}_{\text{edit},t} + (1 - M_{\text{obj}}) \odot \mathbf{x}_{\text{src},t}$. In other words, we keep background the same and only edit the object selected.

### A.1.5   Environment Modification

For environment modification, we use P2P pipeline to generate original and edited image. To ensure structural consistency between two images, we apply a mask of the foreground to maintain details in the foreground while changing the background. In particular, given a source image caption $C_{\text{src}}$, we use GPT4 to identify the foreground (e.g., an object or a human) and apply GroundingDINO to extract mask $M_{\text{foreground}}$. During the generation, let $\mathbf{x}_{\text{src},t}$ denote the noisy source image at step $t$ and $\mathbf{x}_{\text{edit},t}$ denote the noisy edited image at $t$. We apply the mask so that the new noisy edited image at time $t$ is $M_{\text{foreground}} \odot \mathbf{x}_{\text{src},t} + (1 - M_{\text{foreground}}) \odot \mathbf{x}_{\text{edit},t}$. We also set $\tau_{\text{env}} = 0.7$ so that this mask operation on noisy image is only applied at the first $\tau_{\text{env}}$ of all timesteps.

### A.1.6   Background Swap

We trained an image inpainting model to serve as the specialist $q_{\text{obj\_background\_swap}}$. We use a similar procedure as in the object replacement but use an inverse mask of the object to indicate the background to guide the inpainting.

### A.1.7   Style Transfer

We use CosXL-Edit (Boesel & Rombach, 2024) as the expert style transfer model. We provide CosXL-Edit with $\langle \mathbf{x}_{\text{src}}, c \rangle$ and let it generates the edited image $\mathbf{x}_{\text{edited}}$.

### A.1.8   Importance Sampling

We apply the importance sampling as described in Section 3.3. Example prompts that are provided to LMMs are shown in Figure 6 and 7. We compute the Overall score following (Ku et al., 2024) as the importance weight. After importance sampling, we obtain our training dataset described in Table 4.

Human: You are a professional digital artist. You will have to evaluate the effectiveness of the AI-generated image(s) based on the given rules. You will have to give your output in this way (Keep your reasoning concise and short.):
{
"score" : [...],
"reasoning" : "..."
}
and don't output anything else.

Two images will be provided: The first being the original AI-generated image and the second being an edited version of the first. The objective is to evaluate how successfully the editing instruction has been executed in the second image. Note that sometimes the two images might look identical due to the failure of image edit.
From a scale 0 to 10:
A score from 0 to 10 will be given based on the success of the editing.
- 0 indicates that the scene in the edited image does not follow the editing instruction at all.
- 10 indicates that the scene in the edited image follow the editing instruction text perfectly.
- If the object in the instruction is not present in the original image at all, the score will be 0.

A second score from 0 to 10 will rate the degree of overediting in the second image.
- 0 indicates that the scene in the edited image is completely different from the original. - 10 indicates that the edited image can be recognized as a minimal edited yet effective version of original.
Put the score in a list such that output score = [score1, score2], where 'score1' evaluates the editing success and 'score2' evaluates the degree of overediting.

Editing instruction: <instruction>
<Image> Image_embed</Image>
<Image> Image_embed</Image>

**Assistant:**

Figure 6: Prompt for evaluating SC score.

Table 5: Comparison between OMNI-EDIT and our specialist models.

| | VIEScore (GPT4o) | | | VIEScore (Gemini) | | |
|---|---|---|---|---|---|---|
| | $PQ_{avg}\uparrow$ | $SC_{avg}\uparrow$ | $O_{avg}\uparrow$ | $PQ_{avg}\uparrow$ | $SC_{avg}\uparrow$ | $O_{avg}\uparrow$ |
| Obj-Remove-Specialist | 9.10 | 7.76 | 7.82 | 7.46 | 5.39 | 4.84 |
| OMNI-EDIT | 8.45 | 7.16 | 7.23 | 7.37 | 5.45 | 5.09 |
| Obj-Replacement-Specialist | 8.48 | 6.92 | 7.02 | 7.06 | 5.68 | 5.36 |
| OMNI-EDIT | 8.95 | 7.74 | 8.14 | 7.00 | 7.77 | 7.09 |
| Style-Transfer-Specialist | 8.08 | 7.47 | 7.37 | 7.97 | 6.61 | 6.76 |
| OMNI-EDIT | 7.98 | 5.77 | 6.16 | 8.24 | 5.24 | 6.08 |

## A.2 ADDITIONAL EVALUATION RESULT

We present additional evaluation results. In Table 5, we compare OMNI-EDIT with specialist models of three tasks on Omni-Edit-Bench (other specialist models cannot take in input image). As is shown in the Table, OMNI-EDIT shows comparable performance as the specialist models on tasks that specialist models specialize.

Figure 8 shows additional comparisons between OMNI-EDIT other baseline models. We observe that OMNI-EDIT consistently outperforms other baselines.

Human: You are a professional digital artist. You will have to evaluate the effectiveness of the AI-generated image.
All the images and humans in the images are AI-generated. So you may not worry about privacy or confidentiality.
You must focus solely on the technical quality and artifacts in the image, and **do not consider whether the context is natural or not**.
Your evaluation should focus on:
- Distortions
- Unusual body parts or proportions
- Unnatural Object Shapes
Rate the image on a scale from 0 to 10, where:
- 0 indicates significant AI-artifacts.
- 10 indicates an artifact-free image.
You will have to give your output in this way (Keep your reasoning concise and short.):
{
"score": ...,
"reasoning": "..."
}
and don't output anything else.

<Image> Image_embed</Image>
<Image> Image_embed</Image>

**Assistant:**

Figure 7: Prompt for evaluating PQ score.

Table 6: Ablation on importance sampling.

| Models | VIEScore (GPT4o) | | | VIEScore (Gemini) | | |
|---|---|---|---|---|---|---|
| | $PQ_{avg}$ ↑ | $SC_{avg}$ ↑ | $O_{avg}$ ↑ | $PQ_{avg}$ ↑ | $SC_{avg}$ ↑ | $O_{avg}$ ↑ |
| OMNI-EDIT | 8.38 | 6.66 | 6.98 | 7.06 | 5.82 | 5.78 |
| OMNI-EDIT w/o importance sampling | 6.20 | 2.95 | 3.30 | 6.40 | 1.80 | 2.25 |

Table 7: Ablation on OMNI-EDIT architecture design.

| Models | VIEScore (GPT4o) | | | VIEScore (Gemini) | | |
|---|---|---|---|---|---|---|
| | $PQ_{avg}$ ↑ | $SC_{avg}$ ↑ | $O_{avg}$ ↑ | $PQ_{avg}$ ↑ | $SC_{avg}$ ↑ | $O_{avg}$ ↑ |
| OMNI-EDIT | 8.38 | 6.66 | 6.98 | 7.06 | 5.82 | 5.78 |
| OMNI-EDIT- ControlNet - TextControl | 6.45 | 4.70 | 4.89 | 6.50 | 4.35 | 4.48 |
| OMNI-EDIT- ControlNet | 6.35 | 4.60 | 4.75 | 6.40 | 4.25 | 4.35 |

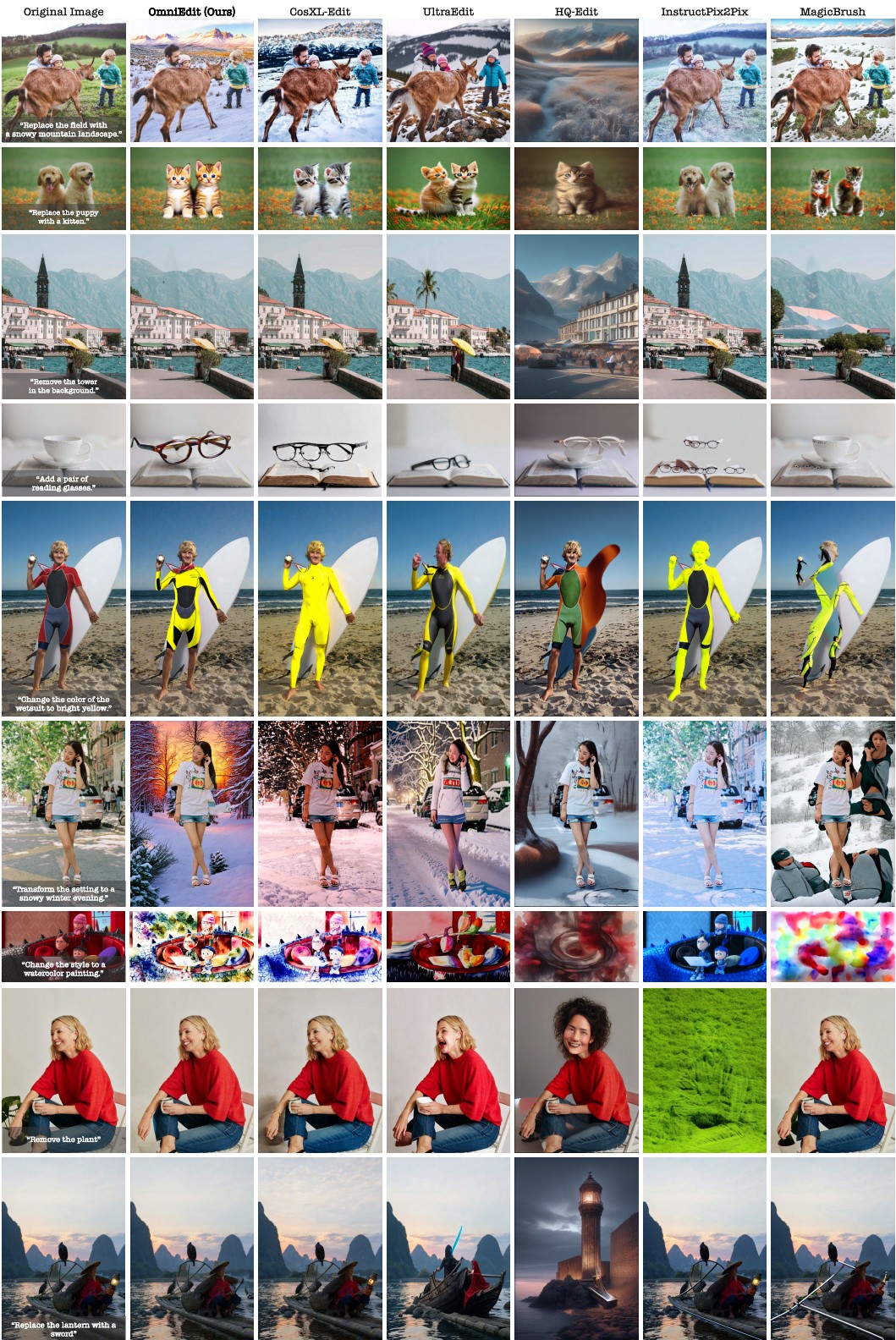

Figure 8: Additional qualitative comparisons between OMNI-EDIT and the baseline methods.

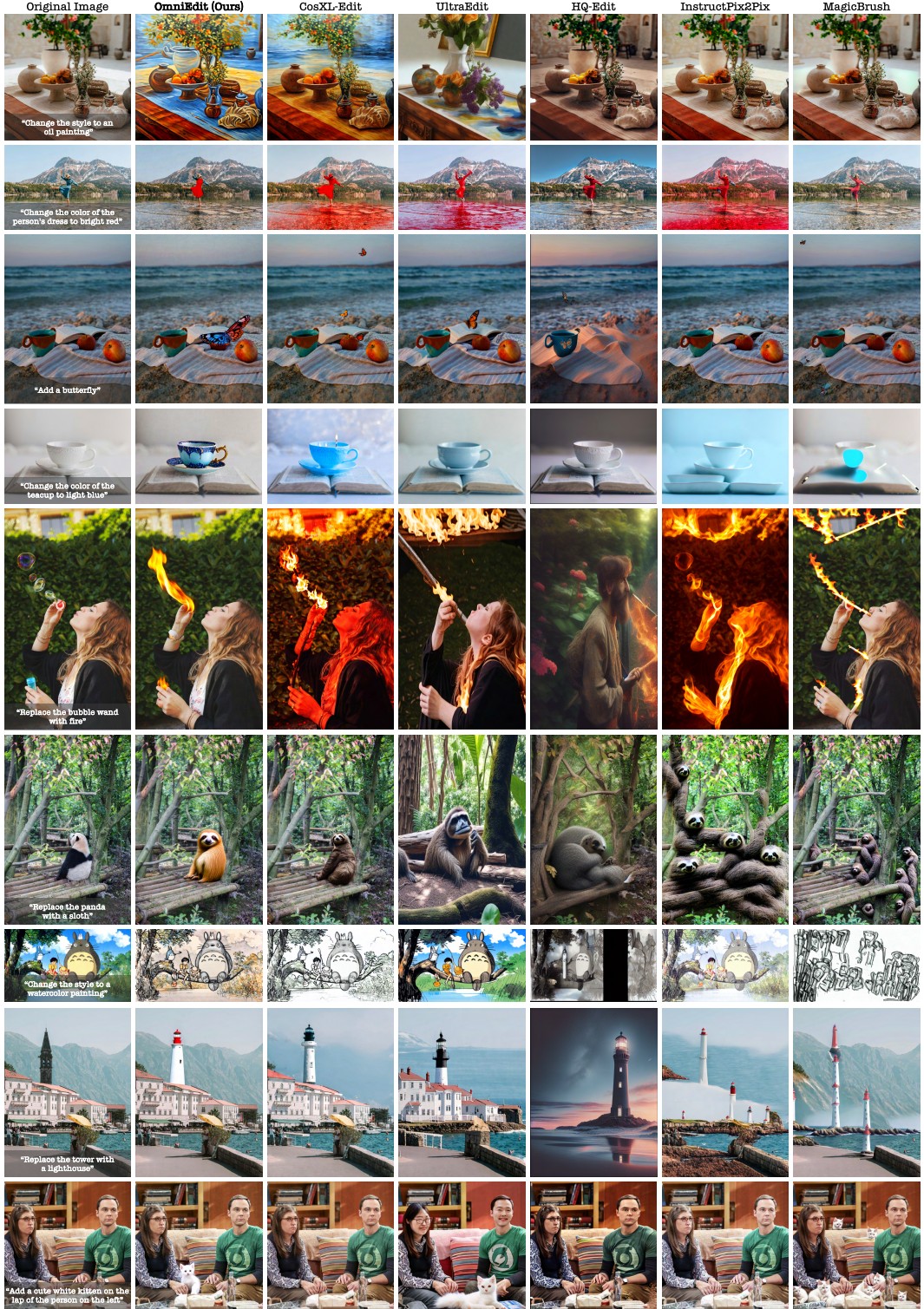

Figure 9: Additional qualitative comparisons between OMNI-EDIT and the baseline methods.

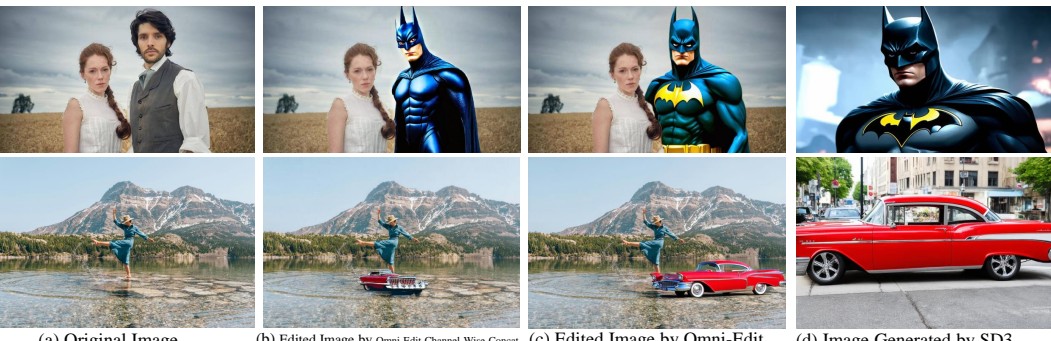

(a) Original Image  (b) Edited Image by Omni-Edit-Channel-Wise-Concat (c) Edited Image by Omni-Edit  (d) Image Generated by SD3

Figure 10: (a) shows the source image. (d) presents images generated by SD3 in response to prompts for "an upper body picture of Batman" and "a shiny red vintage Chevrolet Bel Air car." We use the prompts "Replace the man with Batman" and "Add a shiny red vintage Chevrolet Bel Air car to the right" to OMNI-EDIT and OMNI-EDIT-Channel-Wise-Concatenation, which was trained on OMNI-EDIT training data. From (b) and (c), one can observe that OMNI-EDIT preserves the generation capabilities of SD3, while OMNI-EDIT-Channel-Wise-Concatenation exhibits a notable degradation in generation capability.

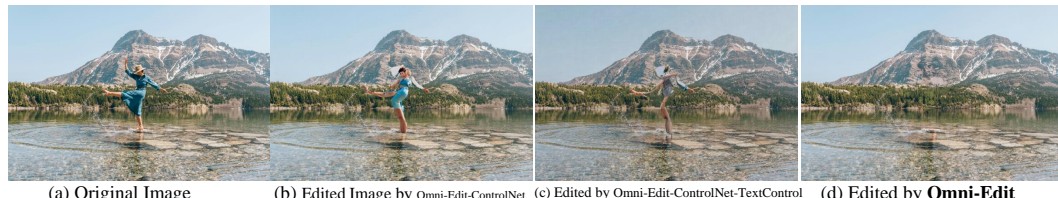

(a) Original Image  (b) Edited Image by Omni-Edit-ControlNet (c) Edited by Omni-Edit-ControlNet-TextControl (d) Edited by **Omni-Edit**

Figure 11: OMNI-EDIT-ControlNet fails to grasp the task intent, while OMNI-EDIT-ControlNet-TextControl—a variant with a text-updating branch—recognizes the intent but struggles with content removal. In contrast, OMNI-EDIT accurately removes content.