# OpenReview forum: "OmniEdit: Building Image Editing Generalist Models Through Specialist Supervision"
_ICLR.cc/2025/Conference — ICLR 2025 Poster_

### Official Review · Reviewer_QAfp · 2024-10-18

**Soundness:** 2
**Presentation:** 3
**Contribution:** 2
**Rating:** 6
**Confidence:** 3

**Summary:**

This paper builds an instruction-based image editing method using synthetic data from 7 specialists. The major contributions include: 1) Curated an (image, editing instruction) dataset for seven editing tasks using the importance sampling strategy; 2) Proposed a ControlNet-like image-editing architecture to achieve these tasks. Experiments show that Omni-Edit outperforms existing approaches after training the proposed model on the curated dataset.

**Strengths:**

1) The curated dataset greatly expands the instruction-based image editing datasets with multiple specialists.

2) The paper standardizes the procedure to automatically generate the trainig data for 7 image editing tasks.

3) The proposed method performs well on both automatic metrics and user studies.

**Weaknesses:**

1) The ablations are not comprehensive. Since the authors claim the new dataset and their model are two separate contributions, they could compare with existing methods trained on their dataset and their method trained on the existing dataset to validate each contribution. For example, the authors can report the results of CosXL-Edit trained on Omni-Edit-Bench.

2) L433 states that while prior methods struggle to perform editing for non-squared images, the proposed method can handle multiple resolutions. However, only two visualized examples are provided in the paper, which is not convincing. I suggest the authors report quantitative results on non-squared images.

**Questions:**

1) Will the dataset be released?

2) When constructing the training data for object removal, how to ensure the generated content in the removal area will be background instead of a new object?

3) Could the authros provide visualized examples of (src image, tgt image, instruction) in the training set for each task?

4) In Tab. 1, It is unclear how different methods are evaluated on their image editing capabilities. Were user studies conducted to get the star ratings? Or are they rated by some automatic metrics?

5) MagicBrush is a human-annotated dataset that also includes 7 tasks. How does the data quality of the synthetic dataset introduced in this paper compare with MagicBrush? Is there a quantitative evaluation?

---

> ### Author Response · Authors · 2024-11-30
> **Response to Reviewer QAfp (1/2)**
>
> Dear reviewer `QAfp`,
>
> We sincerely appreciate the time and effort you have dedicated to reviewing our work and providing valuable feedback. Below, we are pleased to share our detailed responses to your comments.
>
> **W1: More ablations**
>
> We thank the reviewer for raising this important question and have conducted additional ablation studies to address these concerns. Below, we provide three additional sets of experiments to evaluate our dataset and model contributions comprehensively.
>
> CosXL-Edit only releases its model weights, while its training data, data generation process, and training details remain undisclosed and are not described in any paper. Moreover, as noted by the [community](https://huggingface.co/stabilityai/cosxl/discussions/5), there is no guidance on how to train CosXL-Edit, making it difficult for us to train CosXL-Edit on our dataset from scratch.
> Instead, we designed an alternative experiment: we controlled the model. We use SDXL, which shares the same architecture as CosXL-Edit, we train SDXL using channel-wise concatenation (same design employed by CosXL-Edit) on two datasets: our OMNI-EDIT dataset and the best available open-source dataset: UltraEdit 3M. The evaluation results on OMNI-EDIT-Bench are summarized below:
>
> |              |Gemini(SC↑)|Gemini(PQ↑)|Gemini(Overall)| GPT4o(SC↑)| GPT4o(PQ↑)| GPT4o(Overall)|
> |--------------|---------|---------|------|---------|---------|------|
> | UltraEdit(SDXL)      | 6.47       | 4.65        | 4.88        | 6.52    |   4.30  | 4.46  |
> | CosXL-Edit           | **8.34**   | 5.81        | **6.00**    | **7.01**|   4.90  | 4.81  |
> | OmniEdit(SDXL)       | 8.23       | **5.88**    | 5.97        | 6.91    |   **5.12**  | **4.82** |
>
> Key Findings: Our results show that SDXL trained on OMNI-EDIT data achieves comparable performance to CosXL-Edit, despite CosXL as the backbone having superior Image Generation capabilities compared to SDXL, indicating that the quality of our training dataset is at least on par with the undisclosed data used by CosXL-Edit. Furthermore, OmniEdit(SDXL) outperforms UltraEdit(SDXL), demonstrating that our dataset significantly surpasses the quality of the best available open-source alternatives.
>
>
> To further highlight the quality of the OMNI-EDIT dataset, we trained another model, SD3, using channel-wise concatenation on the UltraEdit dataset and OMNI-EDIT dataset, respectively. The evaluation results are presented below:
> |              |Gemini(SC↑)|Gemini(PQ↑)|Gemini(Overall)| GPT4o(SC↑)| GPT4o(PQ↑)| GPT4o(Overall)|
> |--------------|---------|---------|------|---------|---------|------|
> | UltraEdit(SD3)       | 6.44       | 4.66        | 4.86        | 6.49    | 4.33        | 4.45         |
> | OmniEdit(SD3)        | **8.37**       | **5.71**        | **5.96**        | **7.06**    | **4.98**        | **4.82** |
>
> Key Findings: SD3 trained on OMNI-EDIT data significantly outperforms SD3 trained on UltraEdit data across all evaluation metrics, further validating the superior quality of our dataset.
>
> Third, to evaluate the effectiveness of the proposed EditNet architecture. We provide a comparison between SD3 channel-wise concatenation and SD3-EditNet.
> We show that SD3 EditNet significantly improved the generated content quality as indicated by the high PQ scores
> |       |Gemini(SC↑)|Gemini(PQ↑)|Gemini(Overall)| GPT4o(SC↑)| GPT4o(PQ↑)| GPT4o(Overall)|
> |--------------|---------|---------|------|---------|---------|------|
> | OmniEdit(SD3)        | 8.37       | 5.71        | 5.96        | **7.06**   | 4.98        | 4.82         |
> | OmniEdit(SD3-EditNet)| **8.38**   | **6.66**    | **6.98**    | **7.06**    | **5.82**        | **5.78**  |
>
>
> **W2: Visualization Results on non-squared images**
>
> We present visualization samples for all models on the OMNI-EDIT-Bench, as shown in [Figures 16](https://github.com/OmniEditRebuttal/OmniEditRebuttal/blob/main/Figure%2016.png), [Figures 17](https://github.com/OmniEditRebuttal/OmniEditRebuttal/blob/main/Figure%2017.png), [Figure 18](https://github.com/OmniEditRebuttal/OmniEditRebuttal/blob/main/Figure%2018.png), [Figure 19](https://github.com/OmniEditRebuttal/OmniEditRebuttal/blob/main/Figure%2019.png), [Figure 20](https://github.com/OmniEditRebuttal/OmniEditRebuttal/blob/main/Figure%2020.png) and [Figure 21](https://github.com/OmniEditRebuttal/OmniEditRebuttal/blob/main/Figure%2021.png). All visualizations were generated using default configuration and default negative prompts. No cherry-picking was performed, ensuring the results objectively reflect the models' average capabilities.
>
> **Key Observations**: OMNI-EDIT precisely follows instructions while preserving the original image's fidelity. In contrast, **all other baselines introduce undesired modifications, either altering regions that should remain unchanged or changing the overall color tone of the image**

---

> > ### Author Response · Authors · 2024-11-30
> > **Response to Reviewer QAfp (2/2)**
> >
> > **Q1: Will the dataset be released**
> >
> > We confirm that we will release our dataset and we have provided a publicly available random sampled subset for easy visualization here.  [OmniEdit(training set)](https://huggingface.co/datasets/OmniEditRebuttal/training_set_samples)
> >
> > **Q2: For object remover, how to ensure the generated content in the removal area will be background instead of a new object**
> >
> > We have two novel designs for the remover expert.
> > 1. Unique Training Process Tailored to Object Removal: During training, the inpainter is specifically trained to reconstruct masked regions from random strokes instead of object masks. This encourages the model to generate realistic background textures rather than introducing new content during inference.
> > 2. Guessing Background Content Using LMMs: Instead of feeding an empty prompt, we utilize LMMs to predict the likely background content after object removal. This further reduces the likelihood of generating new content.
> >
> > Last but not least, our novel distilled LMM filter serves as a reliable scoring function that can effectively identify and filter out rare failure cases.
> >
> > We further provide a qualitative comparison between our removal experts and SoTA remover in [Figure 8](https://github.com/OmniEditRebuttal/OmniEditRebuttal/blob/main/Figure%208.png). SoTA baseline remover often introduces new content or results in strong artifacts.
> >
> > **Q3: Visualization of training set**
> >
> > Yes in [OmniEdit(training set)](https://huggingface.co/datasets/OmniEditRebuttal/training_set_samples), we visualize a random sampled subset.
> >
> > **Q4:  How different methods are evaluated on their image editing capabilities in Table 1**
> >
> > In our preliminary studies, we curated 50 prompts spanning seven tasks to conduct a human evaluation of the baselines. We enlisted 20 raters to assess the results, assigning star ratings on a scale from 0 to 3 based on the quality of the edits.
> >
> > **Q5: Data Quality Comparison with Human-annotated MagicBrush**
> >
> > We compare our dataset with MagicBrush using VIEScors generated by GPT-4o and Gemini 1.5 Pro, using the same procedure used in the paper on 1000 randomly sampled examples.
> >
> > | VIEScore (GPT4o) | SC ↑ | PQ ↑ | Overall  |
> > |--------------|----|------|----|
> > | MagicBrush  | 7.01 |	7.98 |	6.75
> > | Ours	 | 9.15	| 9.48 |	9.20
> >
> > | VIEScore (Gemini)| SC ↑ | PQ ↑ | Overall  |
> > |--------------|---------|---------|------|
> > | MagicBrush   | 7.09    | 8.11    | 6.78 |
> > | Ours         | 9.12    | 9.45    | 9.17 |
> >
> > Results show that the OmniEdit dataset achieves significantly higher scores across all metrics.

---

> ### Author Response · Authors · 2024-12-02
>
> Dear Reviewer QAfp
>
> Thank you for your insightful feedback and for recognizing the uniqueness of our contributions.
> In this rebuttal, we have provided detailed responses to address the questions and concerns raised. As the rebuttal period is drawing to a close, we would greatly appreciate your feedback on our responses to the reviews!
> Thank you once again for your time and thoughtful review!
>
> Authors

---

### Official Review · Reviewer_dYun · 2024-10-29

**Soundness:** 3
**Presentation:** 3
**Contribution:** 3
**Rating:** 6
**Confidence:** 4

**Summary:**

The paper introduces an effective editing image training data generation process and a new model structure EditNet for general image editing of various image resolutions. The paper proposes to train and utilize seven different specialist image editing models to generate training data and use LLM to filter good-quality images, the paper also proposes a modified model backbone for general image editing capability training. The proposed method is trained on the OMNI-EDIT training dataset and experiments are conducted on the proposed OMNI-EDIT-Bench.

**Strengths:**

1. The proposed training data generation process using seven different specialist models and a better filtering method is novel and reasonable.
2. The modification from simple ControlNet + DiT to the proposed EditNet especially on accessing intermediate presentations and updating text presentations is interesting.
3. The results are strong in both quantitative metrics and qualitative assessments."

**Weaknesses:**

1. I would recommend updating Figure 3, the color of the frozen DiT model is near the same as the white background making readers difficult to comprehend.
2. It would be better if the paper could also include a walkthrough caption other than the only training pipeline image in Figure 2.

**Questions:**

1. I have a question about the generalization ability of the OMNI-EDIT. Since the seven specialist models are trained on their pre-defined tasks, will OMNI-EDIT have the ability to generalize to editing instructions beyond these seven (for example move a person from right to left)?
2. Ideally the specialist model should generate close to perfect edited data, but in real scenarios, for some tasks, a trained specialist model may still not generate great data. Do you encounter this situation where a large portion of the generated data are filtered leading to a small proportion in the final training dataset? Will this data unbalancing affect the model's performance?
3. Can we have some visualization examples of your proposed training data and benchmark?

---

> ### Author Response · Authors · 2024-11-30
> **Response to Reviewer dYun (1/1)**
>
> Dear reviewer `dYun`,
>
> We sincerely appreciate the time and effort you have dedicated to reviewing our work and providing valuable feedback. Below, we are pleased to share our detailed responses to your comments.
>
> **W1,W2,Q3: Update Figure 3, walkthrough for Figure 2, and visualization of training data and benchmark**
> Thanks for your suggestion. We will update our the figure in the final version and we provide the updated [Figure 2](https://github.com/OmniEditRebuttal/OmniEditRebuttal/blob/main/Figure%202.png).
> We also provide a visualization of a randomly sampled training subset in
>  [OmniEdit(training set)](https://huggingface.co/datasets/OmniEditRebuttal/training_set_samples)
>
> We also provide an visualization samples for various models on the OMNI-EDIT-Bench, as shown in [Figures 16](https://github.com/OmniEditRebuttal/OmniEditRebuttal/blob/main/Figure%2016.png), [Figures 17](https://github.com/OmniEditRebuttal/OmniEditRebuttal/blob/main/Figure%2017.png), [Figure 18](https://github.com/OmniEditRebuttal/OmniEditRebuttal/blob/main/Figure%2018.png), [Figure 19](https://github.com/OmniEditRebuttal/OmniEditRebuttal/blob/main/Figure%2019.png), [Figure 20](https://github.com/OmniEditRebuttal/OmniEditRebuttal/blob/main/Figure%2020.png) and [Figure 21](https://github.com/OmniEditRebuttal/OmniEditRebuttal/blob/main/Figure%2021.png). All visualizations were generated using default configuration, and default negative prompts. No cherry-picking was performed, ensuring the results objectively reflect the models' average capabilities.
>
> **Key Observations**: OMNI-EDIT precisely follows instructions while preserving the original image's fidelity. In contrast, **all other baselines introduce undesired modifications, either altering regions that should remain unchanged or changing the overall color tone of the image**
>
> **Q1:  Will OMNI-EDIT have the ability to generalize editing instructions beyond such as (for example moving a person from right to left)**
>
> This is a very interesting question. Currently, OMNI-EDIT has not demonstrated the ability to generalize to tasks like moving an object's position, which was not present in its training set. We hypothesize that this limitation arises because such instructions fall outside the domain of the text-to-image generation backbone. Moreover, there are no demonstrations for this specific type of task in the training data. Consequently, the model cannot generalize to these instructions. Scaling up the training data and increasing the model size could potentially enable emergent abilities. We plan to explore these possibilities in future work.
>
> **Q2: For some tasks, a trained specialist model may still not generate great data. Do you encounter this situation where a large portion of the generated data are filtered leading to a small proportion in the final training dataset? Will this data unbalancing affect the model's performance?**
>
> Thank you for pointing this out.
>
> Yes, we conducted empirical analyses on individual experts before generating the training data. We found that tasks like object swapping and removal have lower generation success rates from the expert model—around 1/5 for producing high-quality examples. In contrast, other tasks achieve a higher success rate of approximately 1/3. To address this, we generated a larger volume of raw data for tasks with lower success rates, ensuring that sufficient high-quality examples remained after filtering. This approach helps ensure that the final dataset is not unbalanced.
>
> We also conducted empirical analyses to assess the learning difficulty of individual tasks by training the model on a single task at a time. From this analysis, we observed that learning difficulty varies significantly across tasks. In general, tasks such as object swapping, addition, and removal are more challenging for the model to learn, while style transfer tasks are the easiest. Other tasks fall somewhere in between.
>
> Based on these findings, we designed the final dataset to prioritize balance in task difficulty rather than the sheer quantity of examples. As detailed in [Table 5](https://github.com/OmniEditRebuttal/OmniEditRebuttal/blob/main/Table%205.png) of our paper, the final training dataset includes approximately 100K–150K samples for harder tasks like object swapping, addition, and removal. For intermediate tasks, we included around 50K samples, while style transfer tasks were limited to 25K samples. This distribution ensures that the model avoids overfitting to easier tasks while optimizing performance on harder ones. We found that this data configuration promotes stable performance across tasks, prevents overfitting, and supports robust generalization.

---

> ### Author Response · Authors · 2024-12-02
>
> Dear Reviewer dYun
>
> We sincerely appreciate your insightful feedback and for recognizing the uniqueness of our contributions.
> In this rebuttal, we have provided detailed responses to address the questions and concerns raised. As the rebuttal period is drawing to a close, we would greatly appreciate your feedback on our responses to the reviews!
> Thank you once again for your time and thoughtful review!
>
> Authors

---

> ### Comment · Reviewer_dYun · 2024-12-02
>
> I appreciate the author's response. Most of my concerns have been addressed, though the generalization question still remains a partial concern for real-life usage. However, as the method demonstrates its novel formulation and great performance in various image editing sub-tasks, I still lean towards recommending an accept as a reviewer, therefore I will keep my current rating of marginal acceptance.

---

### Official Review · Reviewer_KWLC · 2024-11-02

**Soundness:** 3
**Presentation:** 3
**Contribution:** 3
**Rating:** 6
**Confidence:** 3

**Summary:**

This paper presents a new multi-task image editing model. The key contributions of this paper includes (1) framework for learning generalist from data generated from a set of specialist models, (2) scoring of data generated from specialist using a finetuned VLM model, (3) control-net like architecture with DiT. Authors conducted experiments to compare multiple recent methods and did a nice ablation study analyzing components of the proposed model.

**Strengths:**

1. The paper is well-written and easy to read.
2. The proposed method pipeline sound reasonable.
2. Experiments are reasonable with comparisons to many recent methods, and good ablation study.

**Weaknesses:**

1. Lacks novelty. The most significant novel idea of this paper is to train specialist editing models to generate training data for training unified generalist editing model. This idea is quite straightforward and I think many existing multi-task generation models mixes data from different sources, either existing data, or data generated by some models (can be specialist) for joint training from multiple data sources.
2. Results quality. The visual results are not impressive (shadow/reflection is left after removal, inpainting/visual artifact in several results, cartoonish object generated given photo-realistic background, change color instruction lead to texture changes, etc.)

**Questions:**

1. The authors rely on GPT 4o knowledge to build the scoring model. How robust is the scoring model with InternVL in ranking the quality of the data point, how's the accuracy level in general?
2. Could the scoring model be further improved by finetuning with human labeled GT data?
3. It seems the inpainting specialist model cannot handle shadow/reflection. Any idea to improve it?
4. Authors rely on half synthetic data for training, is the trained model affected by the inpainting artifact caused by the synthetic generation?

---

> ### Author Response · Authors · 2024-11-30
> **Response to Reviewer KWLC (1/2)**
>
> Dear reviewer `KWLC`,
>
> We sincerely appreciate the time and effort you have dedicated to reviewing our work and providing valuable feedback. Below, we are pleased to share our detailed responses to your comments.
>
> **W1: Lacks novelty**
>
> We respond to the novelty concern in our  [General Response: The idea of mixed data training is used in other domain](https://openreview.net/forum?id=Hlm0cga0sv&noteId=RFjwXguSIX).
> While we agree that many existing multi-task generation models leverage data from various pre-existing sources, we want to emphasize that datasets in the image editing domain are significantly lacking. we also want to emphasize that creating such datasets is inherently more challenging compared to other vision tasks, and our work directly tackles this critical gap.
> To bridge this gap, we are the first to formalize the data generation pipeline using Importance Sampling with Ensemble Models, as detailed in our general response. Our implementation of the importance sampling function and the development of specialist models are both novel and non-trivial contributions.
> Furthermore, utilizing our newly created dataset, we are the first to demonstrate the effectiveness of a multi-specialist-to-generalist framework in the context of the image editing domain.
>
>
> **W2: Results quality**
>
> Thank you for pointing this out. We did not perform any cherry-picking for the visualizations presented in the paper.  The results were randomly selected to provide readers with an objective understanding of the model's average capabilities. As such, there are examples where OMNI-EDIT did not perform perfectly or where results could be further improved by tuning configurations. However, on average, OMNI-EDIT still significantly outperforms all baselines, including CosXL-Edit, whose training distribution and method remain unclear. Although OMNI-EDIT is not perfect, it represents a promising approach for generating high-quality image editing data and paves the way for future work.
>
> Regarding specific concerns, including "shadows left after removal," "cartoonish objects generated in photo-realistic backgrounds," or "texture changes when altering color," we identified that these issues arise from using excessively high Classifier-Free Guidance (CFG) scores. By reducing the CFG score to 7, we observed notable improvements in these areas. For "object removal," a CFG 4 will make the artifacts unnoticeable.
>
> **To further address the concerns, we provide additional visualization samples for various models on the OMNI-EDIT-Bench. These samples are available in the following figures**:
>  [Figures 16](https://github.com/OmniEditRebuttal/OmniEditRebuttal/blob/main/Figure%2016.png), [Figures 17](https://github.com/OmniEditRebuttal/OmniEditRebuttal/blob/main/Figure%2017.png), [Figure 18](https://github.com/OmniEditRebuttal/OmniEditRebuttal/blob/main/Figure%2018.png), [Figure 19](https://github.com/OmniEditRebuttal/OmniEditRebuttal/blob/main/Figure%2019.png), [Figure 20](https://github.com/OmniEditRebuttal/OmniEditRebuttal/blob/main/Figure%2020.png) and [Figure 21](https://github.com/OmniEditRebuttal/OmniEditRebuttal/blob/main/Figure%2021.png).
>
> Key Observations: OMNI-EDIT precisely follows instructions while preserving the original image's fidelity. In contrast, **all other baselines introduce undesired modifications, either altering regions that should remain unchanged or changing the overall color tone of the image
>
> **Q1: Robustness of distilled InternVL2**
>
> We conduct additional experiments to test the robustness of distilled InternVL2 in our [General Response: Quality and Robustness of scores generated by LMMs](https://openreview.net/forum?id=Hlm0cga0sv&noteId=9swCOd6l0z).
>
> **Q2: Can the scoring model be further improved by ft with human labels**
>
> We believe that the scoring model can be further improved with human-labeled ground-truth data. However, the cost of creating such a dataset would be large. On the other hand, in our [General Response: Quality and Robustness of scores generated by LMMs](https://openreview.net/forum?id=Hlm0cga0sv&noteId=9swCOd6l0z), we show that distilled InternVL2 aligns well with human ground truth scores and serves as a reliable scoring function.
>
> **Q3: The inpainting specialist model cannot handle shadow/reflection**
>
> During inference, we utilize DINO and Grounded-SAM to generate masks for the inpainting specialist model. The inpainting specialist is capable of handling shadows and reflections, provided that these elements are included in the mask. For an example, refer to [Figure 22](https://github.com/OmniEditRebuttal/OmniEditRebuttal/blob/main/Figure%2022.png), in this example, Grounded-SAM successfully identify the reflections, then the inpainter expert can produce good data.
> So the primary challenge lies in the mask predictor's performance. To enhance the pipeline, integrating an end-to-end mask predictor that can effectively detect shadows and reflection according to the object may be necessary.

---

> ### Author Response · Authors · 2024-11-30
> **Response to Reviewer KWLC (2/2)**
>
> **Q4: Is the model affected by inpainting artifact?**
>
> For object swapping, we reverse the roles of the source image and the target image during training. This ensures that the optimization target does not contain artifacts, helping to mitigate the artifacts introduced by the inpainter.
> For both object removal and object swapping, as shown in the visualizations and in [Table 8-10](https://github.com/OmniEditRebuttal/OmniEditRebuttal/blob/main/Table%208-10.png), our model achieves the highest performance. This demonstrates that our proposed training paradigm, along with the LMM filter, effectively eliminates examples with inpainting artifacts.

---

> ### Author Response · Authors · 2024-12-02
>
> Dear Reviewer KWLC,
>
> We sincerely appreciate your insightful feedback and for recognizing the uniqueness of our contributions.
> In this rebuttal, we have provided detailed responses to address the questions and concerns raised. As the rebuttal period is drawing to a close, we would greatly appreciate your feedback on our responses to the reviews!
> Thank you once again for your time and thoughtful review!
>
> Authors

---

> > ### Comment · Reviewer_KWLC · 2024-12-02
> >
> > Thanks for the authors for detailed responses to my review questions. They addressed some of my concerns. I still see the values of this work being published to the community. Hence, I am happy to slightly increase my score.

---

### Official Review · Reviewer_u8AW · 2024-11-03

**Soundness:** 3
**Presentation:** 3
**Contribution:** 2
**Rating:** 5
**Confidence:** 5

**Summary:**

The paper introduces OMNI-EDIT, an advanced image editing model addressing current limitations such as biased training data, noise, and fixed aspect ratios. It uses supervision from multiple specialist models, improved data quality via importance sampling with large multimodal models, and a new architecture called EditNet for better editing performance. OMNI-EDIT supports various aspect ratios and outperforms existing models in evaluations.

**Strengths:**

--The paper introduces OMNI-EDIT to address limited editing capabilities, poor data quality control, and lack of support for varying resolutions, which are critical point in image editing method.

--The use of importance sampling with high-quality data scoring via distilled multimodal models (such as InternVL2) demonstrates a meticulous approach to data curation

--The paper is clearly structured, with each contribution outlined and explained in detail. The authors provide a logical flow that makes it easy for readers to understand how OMNI-EDIT was developed.

**Weaknesses:**

1. The primary concern with this paper is its similarity to a CVPR 2024 paper [1], which proposed a generalist modeling interface that formulates most vision tasks as instruction-based editing. In contrast, the proposed paper is simpler and more direct as it focuses on building a generalist model specifically for editing tasks rather than all vision tasks. The training approach is similar to [1] which also involves generating data using current specialist models. This reduces the contribution of leveraging specialist models to train the generalist model. Additionally, the experimental results lack a comparison with [1], which would be necessary to highlight differences.

2. Although the paper claims that the model supports inputs of varying sizes and resolutions, there does not appear to be a detailed explanation of this in the methodology or experimental sections. This is crucial information, as it raises questions about how this feature is implemented and whether it involves any novel techniques.

3. The training data for Omni-Edit consists of 505K samples, all generated by specialist models and selected using importance sampling. It would be beneficial to discuss whether different training data sizes affect the model’s results.

4. Some specialist models mentioned training an image-inpainting model, but there seems no description of the training data for specialist models.

5. The overall framework is complex, requiring the training of specialist models, a score prediction model for importance sampling, and finally the Omni-Edit model with all the collected data. And such a process makes it difficult to dynamically include more editing tasks.

6. There seems to be unnecessary "5" in Lines 361 and 362.

7. Typo: In Line 246, "for the our editing model".

【1】InstructDiffusion: A Generalist Modeling Interface for Vision Tasks.

**Questions:**

1. How does this paper differentiate itself in terms of contributions from the CVPR 2024 paper [1], given their similar focus on generalist instruction-guided modeling?

2. It would be better to compare the proposed method with the generalist modeling approach in [1].

3. Can the authors provide more detail on how the support for multi-size and multi-resolution inputs was implemented? What specific innovations, if any, were introduced to achieve this?

4. How does the training data size affect the performance of OMNI-EDIT?

5. Could the authors elaborate on the training process and datasets used for the specialist models, such as the image-inpainting model?

**Details Of Ethics Concerns:**

The model might exhibit bias in image edits related to race, gender, or the factors stemmed from the specialist models.

---

> ### Author Response · Authors · 2024-11-30
> **Response to Reviewer u8AW (1/3)**
>
> Dear reviewer `u8AW`,
>
> We sincerely appreciate the time and effort you have dedicated to reviewing our work and providing valuable feedback. Below, we are pleased to share our detailed responses to your comments..
>
> **W1, Q1, Q2: Contribution Differences and Comparison with [1] Instructdiffusion**
>
> **Answer:** Thank you for pointing this out. In our [General Response: The idea of mixed data training is used in other domain](https://openreview.net/forum?id=Hlm0cga0sv&noteId=RFjwXguSIX), we address concerns related to our contributions in detail.
>
> We **differentiate our work from InstructDiffusion** by emphasizing that their approach primarily relies on **pre-existing datasets**. In contrast, we directly work on the method to generate a high-quality image editing dataset.  Specifically, as stated in Section 4.1 (Settings: Training Samples) of the InstructDiffusion paper, their training data consists of 245K samples for keypoint detection and 239K for segmentation, all sourced from pre-existing datasets. For image editing, they use 374K samples from seven existing datasets, supplemented by 51K crawled samples, resulting in a total of 425K image editing samples.
>
> In contrast, our contribution lies in proposing **a novel framework to automatically generate large-scale, high-quality image editing training data**. We leverage innovative LMM-based Importance Sampling and carefully designed Ensemble Models to ensure the generation of high-quality images, as visualized in our dataset [OmniEdit(auto pipeline)](https://huggingface.co/datasets/OmniEditRebuttal/training_set_samples).
>
> Additionally, we provide a **detailed comparison of InstructDiffusion and other baseline models** in  [Table 7](https://github.com/OmniEditRebuttal/OmniEditRebuttal/blob/main/Table%207.png). As demonstrated, our method significantly outperforms InstructDiffusion, highlighting the effectiveness of our proposed framework and dataset and the impact of our contributions.
>
> We further provide diverse **visualization samples** for various models **including InstructDiffusion** on the OMNI-EDIT-Bench, as shown in [Figures 16](https://github.com/OmniEditRebuttal/OmniEditRebuttal/blob/main/Figure%2016.png), [Figures 17](https://github.com/OmniEditRebuttal/OmniEditRebuttal/blob/main/Figure%2017.png), [Figure 18](https://github.com/OmniEditRebuttal/OmniEditRebuttal/blob/main/Figure%2018.png), [Figure 19](https://github.com/OmniEditRebuttal/OmniEditRebuttal/blob/main/Figure%2019.png), [Figure 20](https://github.com/OmniEditRebuttal/OmniEditRebuttal/blob/main/Figure%2020.png) and [Figure 21](https://github.com/OmniEditRebuttal/OmniEditRebuttal/blob/main/Figure%2021.png). This further demonstrates the effectiveness of our framework. All visualizations were generated using default configuration and default negative prompts. No cherry-picking was performed, ensuring the results objectively reflect the models' average capabilities.
>
> **Key Observations**: OMNI-EDIT precisely follows instructions while preserving the original image's fidelity. In contrast, **all other baselines introduce undesired modifications, either altering regions that should remain unchanged or changing the overall color tone of the image**
>
>
> **W2&Q3: Explanation on varying sizes and resolutions**
>
> **Answer:** DiT models typically utilize positional embeddings to encode patch positions, which enable the model to understand spatial relationships. However, generalization across multiple resolutions is not guaranteed if the training data is limited to a single resolution, as is the case with all existing image editing datasets(500x500 row resolution square images). To effectively support multi-size and multi-resolution inputs, it is **crucial to train the model on data that encompasses a variety of sizes and resolutions**, ensuring robust performance across diverse inputs.
>
> Our expert models are trained on the LAION-Aesthetic dataset (details provided in Appendix A.1 and in response to "W4&Q5: Details on training specialist models"), which encompasses a diverse range of aspect ratios and resolutions. This allows the expert models to generate training data across various resolutions and aspect ratios. Specifically, our training dataset includes aspect ratios such as 1:1, 2:3, 3:2, 3:4, 4:3, 9:16, and 16:9. This diversity ensures that our final model OMNI-EDIT is well-equipped to handle image editing tasks effectively across a wide range of resolutions.  From the **visualization samples**, it is evident that all baseline models, except CosXL-Edit (whose training distribution is not released), were trained on low-resolution, noisy datasets, resulting in unintended alterations to regions that should have remained unchanged, and difficulties in handling various aspect ratios.

---

> ### Author Response · Authors · 2024-11-30
> **Response to Reviewer u8AW (2/3)**
>
> **W4&Q5: Details on training specialist models**
>
> **Answer:** Thank you for the suggestion. In response, we have provided a detailed description of the training data and model in Appendix A.1: Training Data Generation Details. Briefly, we use the LAION-Aesthetic dataset as the training data for our inpainting experts. These experts are built upon the BrushNet architecture and are initialized from Juggernaut-XL.
> As discussed in the  [General Response: The idea of mixed data training is used in other domain](https://openreview.net/forum?id=Hlm0cga0sv&noteId=RFjwXguSIX), existing automated methods struggle to reliably generate high-quality data for many tasks, particularly for high-resolution images, where artifacts are more pronounced.
> For example, [Figure 7](https://github.com/OmniEditRebuttal/OmniEditRebuttal/blob/main/Figure%207.png) shows that SoTA inpainter cannot effectively swap the lion with tiger as it will preserve the shape of the original object; [Figure 8](https://github.com/OmniEditRebuttal/OmniEditRebuttal/blob/main/Figure%208.png) shows that SoTA remover often introduce new content or results in strong artifacts; [Figure 9](https://github.com/OmniEditRebuttal/OmniEditRebuttal/blob/main/Figure%209.png) shows that P2P pipeline suffers significant image consistency issue.
> To enhance the editing success rate, we propose innovative training and inference paradigms for each expert, as detailed in Appendix A.1.
> In short, when training the inpainter model for object-swapping tasks, we employ multiple mask augmentations to **mitigate its sensitivity to mask shapes.**
> We train removal experts to inpaint random strokes instead of objects and use LMMs to generate descriptions for the background to **avoid the introduction of new objects**. We apply a meticulously designed masking strategy during the Prompt2Prompt pipeline to highly **increase its consistency**.
>
> **W5: The overall framework is complex. Such a process makes it difficult to dynamically include more editing tasks**.
>
> **Answer:**  The complexity of the proposed framework arises naturally from the challenges inherent in creating high-quality image editing datasets. Achieving a simple yet robust framework to generate image editing data at scale requires addressing the "chicken-and-egg" problem: If such a simple and robust method had already existed, the problem would have been solved by that method alone.
>
> Our framework addresses this gap by leveraging specialist models, which can be easily scaled up during their training to effectively handle individual tasks. This strategy enables the synthesis of diverse, large-scale and high-quality datasets. While this process involves multiple steps (training specialists, importance score prediction, and Omni-Edit model training), each step is essential to achieving state-of-the-art performance in this challenging domain.
>
> Importantly, the framework retains flexibility for dynamically incorporating new editing tasks. To include a new task, one only needs an expert capable of generating data for the task. The score prediction model can then directly filter this newly generated dataset, allowing it to be seamlessly added to the existing training set. This modular approach ensures that the framework remains scalable and adaptable to evolving needs in image editing tasks.
>
> **W6&W7**: Typos
>
> **Answer:** Thank you for pointing out these issues. We have corrected them in the updated PDF.
>
> [1] Geng, Z., Yang, B., Hang, T., Li, C., Gu, S., Zhang, T., ... & Guo, B. (2024). Instructdiffusion: A generalist modeling interface for vision tasks.

---

> > ### Author Response · Authors · 2024-12-02
> > **Response to Reviewer u8AW (3/3)**
> >
> > **W3&Q4: Ablation study on different training sizes**
> >
> > We train another model with 50K data (10% of our whole dataset) and calculate the VIEScore measured by GPT4o and Gemini. We can observe that the dataset size matters as the model trained with less data has a lower performance.
> >
> > | Training size| PQ_avg↑ (GPT4o) | SC_avg↑ (GPT4o) | O_avg↑ (GPT4o) | PQ_avg↑ (Gemini) | SC_avg↑ (Gemini) | O_avg↑ (Gemini) |
> > |--------------|-----------------|-----------------|-------------|--------------|------------------|-----------------|
> > | 50K |  6.32  |  4.73 |  4.92  | 6.61 | 4.19| 4.40 |
> > | Full (505K) | 8.38 |  6.66  | 6.98 | 7.06 |  5.82   | 5.78 |

---

> ### Author Response · Authors · 2024-12-02
>
> Dear Reviewer u8AW
>
> We sincerely appreciate your insightful feedback and for recognizing the uniqueness of our contributions.
> In this rebuttal, we have provided detailed responses to address the questions and concerns raised. As the rebuttal period is drawing to a close, we would greatly appreciate your feedback on our responses to the reviews!
> Thank you once again for your time and thoughtful review!
>
> Authors

---

> ### Author Response · Authors · 2024-12-03
>
> Dear Reviewer u8AW,
>
> We sincerely appreciate your insightful feedback and acknowledgment of the uniqueness of our contributions. In this rebuttal, we have carefully addressed the questions and concerns you raised, providing detailed responses to each point.
>
> As the rebuttal period is nearing its conclusion, we would be deeply grateful for any additional feedback or thoughts you might have on our responses. Your input is invaluable to us, and we thank you once again for your time, effort, and thoughtful review！
>
> Best regards,
>
> The Authors

---

### Official Review · Reviewer_txht · 2024-11-04

**Soundness:** 3
**Presentation:** 3
**Contribution:** 2
**Rating:** 6
**Confidence:** 3

**Summary:**

OMNI-EDIT is an image editing model designed to address the skill imbalance issues present in existing instruction-based image editing methods. It learns by combining supervision from seven different expert models and utilizes InternVL2 for data scoring. Additionally, this approach introduces a diffusion-Transformer architecture named EditNet.

**Strengths:**

1. The authors collected a high-resolution dataset covering seven types of tasks, annotated through multiple expert models, and used InternVL2 to score the dataset.

2. And they introduced EditNet, which enhances image noise control capabilities compared to ControlNet.

3. In the Omni-Edit-Bench benchmark, OMNI-EDIT outperformed existing baseline models across multiple evaluation metrics.

**Weaknesses:**

1. Lack of testing of GPT4-o generated CONFIDENCE SCORE, LLM may be subjective and biased

2. Limitations of the benchmarking dataset: The Omni-Edit-Bench dataset used in the paper is small (only 62 images), which may limit the reliability and generalization of the model evaluation.

**Questions:**

1. Since the dataset is divided into seven types of tasks, would it be possible to categorize the test set similarly? This would allow for a more detailed evaluation of the model's performance across different editing tasks and facilitate an analysis of performance differences and underlying reasons.

2. The relatively small size of the Omni-Edit-Bench dataset may limit the reliability and generalizability of model evaluation. It is recommended to assess performance on additional benchmark datasets. For example, are there results available on the Emu Edit test set?

---

> ### Author Response · Authors · 2024-11-30
> **Response to Reviewer txht (1/2)**
>
> Dear reviewer `txht`,
>
> We sincerely appreciate the time and effort you have dedicated to reviewing our work and providing valuable feedback. Below, we are pleased to share our detailed responses to your comments.
>
> **W1: Lack of testing of GPT4-o generated CONFIDENCE SCORE, LLM may be subjective and biased**
>
> Thank you for pointing this out!
> We presented an additional study in the [General Response: Quality and Robustness of scores generated by LMMs](https://openreview.net/forum?id=Hlm0cga0sv&noteId=9swCOd6l0z), showing that our proposed **InternVL2-Distilled filter** **aligns closely** with **human expert judgments**. This is demonstrated by high precision, recall, and average VIEScore when compared to ground truth labels created by human experts. Moreover, our method significantly outperforms the traditional CLIP-based filter used in prior works, which may also be subjective and biased due to its reliance on the biases inherent in its training distribution [1][2].
>
> We also want to highlight that the VIEScore paper [1] studied the robustness of scores generated by LLMs and traditional metrics across various generation tasks. Their findings show that GPT-4o's VIEScore is the best metric in terms of alignment with human evaluations. This further supports the reliability of LLM-generated scores for complex filtering tasks.
>
> Overall, our results demonstrate the effectiveness of the **proposed LLM filtering method** and **novel scoring-ability distillation method**, which distills GPT-4o's scoring ability into a **small and cheap** model, InternVL2-Distilled, to create a **scalable and accurate filtering**  mechanism. This approach shows great promise for generating **high-quality, large-scale** datasets and highlights its potential for future work.
>
> We acknowledge that the resulting filter may still be subjective and prone to bias. Fine-tuning on diverse, human-labeled datasets could further improve its robustness. However, rigorous testing for LLM bias and subjectivity was beyond the scope of our current work and will be a focus for future research.
>
> [1] Ku, M., Jiang, D., Wei, C., Yue, X., & Chen, W. (2023). Viescore: Towards explainable metrics for conditional image synthesis evaluation.
>
> [2] Discovering and Mitigating Visual Biases Through Keyword Explanation.
>
>
> **W2: Limitation of Omni-Edit-Bench; Q2: Evaluation on additional benchmark dataset**
>
> We thank the reviewer for the suggestion.
> To address the reviewer's concerns, we have conducted an additional evaluation of various models on the Emu Edit test set, as presented in  [Table 15](https://github.com/OmniEditRebuttal/OmniEditRebuttal/blob/main/Table%2015.png).
>
> We would like to highlight several key observations from this evaluation:
>
> * OmniEdit achieved the highest scores on metrics indicating editing quality:
> CLIP-Dir (measuring whether the editing is consistent with the changes described between the captions of the source and target images) and
> CLIP-T (measuring whether the edited image aligns with the target image caption).
> For content preservation metrics, such as CLIP-I and DINO, OmniEdit achieved second-highest scores. MagicBrush achieved the highest scores simply because its tendency to produce identity images with minimal edits, as evidenced by its' low CLIP-Dir and CLIP-T scores.
> * OmniEdit strikes the best balance, producing accurate edits with minimal overediting, which is crucial for achieving both editing precision and content preservation.
>
> We also provide a qualitative comparison between Omni-Edit and Emu edit in [Figure 14](https://github.com/OmniEditRebuttal/OmniEditRebuttal/blob/main/Figure%2014.png).
>
> We would like to provide additional context regarding the scope and design of the Omni-Edit-Bench dataset. Omni-Edit-Bench is the first benchmark that comprises manually selected high-resolution images with high aesthetic scores and a balanced distribution of aspect ratios and domain, closely resembling real-world editing scenarios. In total, the benchmark includes 434 diverse edits.
> In comparison, Emu Edit’s test set, while larger in size, is limited to only 500x500 images and is constrained to the MS-COCO domain. Many images in the datasets are blurry or paired with incorrect captions, which further limits their ability to evaluate instruction-based image editing models in real-world applications. Due to these limitations, we did not evaluate our model on this dataset initially.

---

> ### Author Response · Authors · 2024-11-30
> **Response to Reviewer txht (2/2)**
>
> **Q1: Categorize the test set into seven types of tasks**
>
> **Answer:** Thank you for the suggestion!
> In response, we have categorized the test set into the seven types of tasks as suggested and provided detailed evaluations for each category. This additional analysis is included in [Table 8-10](https://github.com/OmniEditRebuttal/OmniEditRebuttal/blob/main/Table%208-10.png), [Table 11-13](https://github.com/OmniEditRebuttal/OmniEditRebuttal/blob/main/Table%2011-13.png), [Table 14](https://github.com/OmniEditRebuttal/OmniEditRebuttal/blob/main/Table%2014.png).
>
>
> We also provide an **visualization samples for various models on the OMNI-EDIT-Bench**, as shown in [Figures 16](https://github.com/OmniEditRebuttal/OmniEditRebuttal/blob/main/Figure%2016.png), [Figures 17](https://github.com/OmniEditRebuttal/OmniEditRebuttal/blob/main/Figure%2017.png), [Figure 18](https://github.com/OmniEditRebuttal/OmniEditRebuttal/blob/main/Figure%2018.png), [Figure 19](https://github.com/OmniEditRebuttal/OmniEditRebuttal/blob/main/Figure%2019.png), [Figure 20](https://github.com/OmniEditRebuttal/OmniEditRebuttal/blob/main/Figure%2020.png) and [Figure 21](https://github.com/OmniEditRebuttal/OmniEditRebuttal/blob/main/Figure%2021.png). All visualizations were generated using default configuration, and default negative prompts. No cherry-picking was performed, ensuring the results objectively reflect the models' average capabilities.
>
> **Key Observations**: OMNI-EDIT precisely follows instructions while preserving the original image's fidelity. In contrast, **all other baselines introduce undesired modifications, either altering regions that should remain unchanged or changing the overall color tone of the image**

---

> ### Author Response · Authors · 2024-12-02
>
> Dear Reviewer txht,
>
> We sincerely appreciate your insightful feedback and for recognizing the uniqueness of our contributions.
> In this rebuttal, we have provided detailed responses to address the questions and concerns raised. As the rebuttal period is drawing to a close, we would greatly appreciate your feedback on our responses to the reviews!
> Thank you once again for your time and thoughtful review!
>
> Authors

---

> > ### Comment · Reviewer_txht · 2024-12-02
> >
> > Thanks for your rebuttal, most of the doubts have been explained, I will raise my score.

---

### Author Response · Authors · 2024-11-30
**General Response (1/3)**

We appreciate the reviewers for their thoughtful feedback.

We greatly appreciate your recognition of the strengths of our work as follows:

 **Novel Data Generation Method and Novel Filtering Method**

Our Multi-Specialist data generation pipeline "**is novel**"(`R-dYun`)  and **"sounds reasonable"**(`R-KWLC`,`R-dYun`), "**standardizes the procedure** to automatically generate the training data" (`R-QAfp`), addressing "the **critical point** in image editing method" (`R-u8AW`).

Our proposed LLM Filter "**is novel**"(`R-dYun`)  and **"reasonable"**(`R-KWLC`,`R-dYun`), "demonstrates a **meticulous approach** to data curation."(`R-u8AW`), addressing "the **critical point** in image editing method." (`R-u8AW`)

Our curated dataset "**greatly expands** the instruction-based image editing datasets with multiple specialists."  (`R-QAfp`)

Our EditNet architecture design "**enhances image noise control capabilities**"(`R-txht`) and "**is interesting**"(`R-dYun`)

# General Response: Quality and Robustness of scores generated by LMMs (R-txht, R-KWLC)
We provide additional experiments to compare the performance of three filtering approaches: (1) the **CLIP filter**, which has been the primary filtering method in all prior works; (2) our proposed **novel LMM filters**; and (3) evaluations by a **human expert**.

Specifically, we randomly sampled 500 examples from our pre-filtered raw dataset for this analysis.

For **CLIP** score-based filtering, we follow the pipeline from InstructPix2Pix [1] to calculate image-image CLIP similarity, image-caption CLIP similarity for both original and edited images, and directional CLIP similarity. We set the thresholds at 0.75, 0.2, and 0.2 respectively, strictly adhering to the InstructPix2Pix methodology.

For **InterVL2-8B**, **InternVL2-8B-distilled (from-GPT4o)**, **GPT-4o** and four **human experts**, each method was tasked with generating a VIEScore [2] for every sample. We then apply a threshold of overall score $\geq$ 9.0 to select high-quality images.

We treated the selections made by **human experts as the ground truth**. We calculated the **Precision**, **Recall** and **F1-Score** to measure the overlap between the images selected by each method and those selected by the human experts.
Additionally, we calculated the Pearson **correlation** coefficient between each model's predicted scores and the human experts' overall scores. Finally, we ranked the 500 examples based on each model's predicted scores and the human experts' scores to calculate the **Spearman’s rho** correlation.

|Filtering Method         | Recall | Accuracy | F1-Score |  Spearman’s rho    |Correlation|
|-------------------------|--------|----------|----------|---------|-----------|
| Human               | 1.00   |     1.00 | 1.00     |   N/A   |  N/A      |
| GPT-4o                  | 0.67   |     0.96 | 0.79  |  0.68   | 0.77      |
| InternVL2-8B-distilled  | 0.54   | 0.84 | 0.66 |  0.67   | 0.67      |
| InternVL2-8B            | 0.05  | 0.75 |0.09 | -0.08   | -0.12     |
| CLIP | 0.01  | 0.74 | 0.02     | -0.09   | -0.13     |

Our findings are summarized below:
1. GPT-4o and our proposed **InternVL2-8B-distilled** filter serve as the most reliable scoring functions, aligning **very well** with **human** scores, as reflected in their high correlations.
2. **CLIP-based** filtering demonstrates poor alignment with human judgments. While it achieves a 74% Accuracy, this is largely a result of the imbalanced dataset, where negative examples dominate. However, its low Recall and F1-Score of 0 shows that CLIP **fails recognize good training pairs**.
3. The promising results for InternVL2-8B-distilled further validate the effectiveness of our proposed **scoring-ability distillation method** for building a **scalable and accurate** filtering mechanism, which is crucial for generating high-quality, large-scale datasets. In [Figure 15](https://github.com/OmniEditRebuttal/OmniEditRebuttal/blob/main/Figure%2015.png), we show a case comparison between InternVL2-8B's scoring ability before and after distillation.

We believe that our **proposed LMM-based filtering method** will inspire the community to adopt LMMs for image data filtering in the future, particularly for **more complex editing or generation tasks** where traditional methods such as CLIP-based approaches fall short.


### In addition to the quantitative analysis, we provide a randomly sampled **training subset for visualization** at this link: [OmniEdit(auto pipeline)](https://huggingface.co/datasets/OmniEditRebuttal/training_set_samples)
### This allows a direct comparison with **prior works' datasets**, provided at this link:[UltraEdit(auto pipeline)](https://huggingface.co/datasets/BleachNick/UltraEdit_500k), [InstructPix2Pix(auto pipeline)](https://huggingface.co/datasets/timbrooks/instructpix2pix-clip-filtered), [MagicBrush(human-annotated)](https://huggingface.co/datasets/osunlp/MagicBrush).


### All our datasets will be open-sourced!

---

> ### Author Response · Authors · 2024-11-30
> **General Response (2/3)**
>
> # General Response: The idea of mixed data training is used in other domain (R-u8AW, R-KWLC)
> We sincerely thank the reviewer for raising this important point. To address the concern, we emphasize the key distinction between our work and prior studies:
>
> * **Existing studies focus on leveraging multiple pre-existing datasets to train generalist models.** These efforts rely on the availability of high-quality, curated datasets for well-established domains. For instance, tasks like keypoint detection and semantic segmentation have access to large-scale datasets such as COCO and SA-1B.
> * **In contrast, the field of image editing faces a fundamental challenge: the lack of any high-quality datasets to begin with. Our work directly addresses this issue**. As highlighted in the introduction, existing datasets are plagued by a high volume of noise and artifacts, and they are limited to low resolutions and fixed aspect ratios. **This lack of data issue represents a significant bottleneck for image editing domain and is more challenging to overcome** compared to other computer vision tasks (e.g., segmentation, recognition) and generative tasks (e.g., text-to-image generation), where samples can be readily annotated by trained individuals or automated models. On one hand, hiring professional Photoshop experts to manually create large scale datasets is not feasible. On the other hand, existing automated methods struggle to reliably generate high-quality data for many tasks, particularly for high-resolution images, where artifacts are more pronounced. Therefore we carefully design our own automatic pipeline by apply task-specific training or additional engineering design like masking to increase the data generation quality.
>   * [Figure 7](https://github.com/OmniEditRebuttal/OmniEditRebuttal/blob/main/Figure%207.png) shows that SoTA inpainter cannot effectively swap the lion with tiger as it will preserve the shape of the original object;
>   * [Figure 8](https://github.com/OmniEditRebuttal/OmniEditRebuttal/blob/main/Figure%208.png) shows that SoTA remover often introduce new content or results in strong artifacts;
>   * [Figure 9](https://github.com/OmniEditRebuttal/OmniEditRebuttal/blob/main/Figure%209.png) shows that P2P pipeline suffers significant image consistency issue.
>
>
> This makes our work uniquely significant. Here, we emphasize our core contribution:
>
> **Core Contribution 1: A Standardized Approach to Image Editing Data Generation**
>
> As recognized by R-dYun and R-QAfp, our work is the **is the first to mathematically formalize the process of automatically generating image editing training data**. This is achieved through two key components:
>
> 1. Importance Sampling
> 2. Ensemble Models
>
> Together, these components address the core bottleneck of data scarcity in image editing. Our framework, detailed in Equations (5) and (6) and Algorithm 1, provides a novel approach to generating high-quality training data for image editing models.
>
>
> **Core Contribution 2: A Promising Implementation of the Framework**
>
> To demonstrate the potential of our framework, we present its implementation and thoroughly evaluate its performance. Our second core contribution is the exact **design of the importance sampling function** and **expert models**:
>
> (1) Distilled, **Cost-Efficient** LMM for Large-Scale Dataset Filtering:
> **we are the first to introduce a distilled large multimodal model (LMM)** for generating scores and filtering a massive image editing training dataset. Our "scoring-ability distillation" approach is novel, efficient, and reliable. This is reflected in the experiment above and the visualizations provided in the [figure](https://github.com/OmniEditRebuttal/OmniEditRebuttal/blob/main/Figure%2015.png). This approach shows great promise for generating high-quality, large-scale datasets and highlights its potential for future work.
>
> (2) **Effective Design of Expert Models**:
> Each expert model in our framework is carefully engineered to achieve high success rates when dealing with high-resolution images, overcoming the limitations of existing methods.
> * When training the inpainter model for object-swapping tasks, we employ 4 types of mask augmentations to **mitigate its sensitivity to mask shapes.**
> * We train remover experts to reconstruct random strokes instead of objects and use LMMs to generate background content descriptions to **avoid the introduction of new objects**.
> * We apply a meticulously designed masking strategy during the Prompt2Prompt pipeline to highly **increase its consistency**.
>
>
> **Core Contribution 3: Our 505K high-quality, high-resolution, multi-aspect-ratio, multi-task training dataset, along with a 5M pre-filtered dataset, will be made publicly available.**
>
> We believe that the high-quality dataset we have generated will make a significant and direct contribution to the image editing/generation research community.

---

> ### Author Response · Authors · 2024-11-30
> **General Response (3/3)**
>
> **Core Contribution 4: A Novel DiT Architecture for Image Editing**:
>
> Previous approaches to image editing have relied on channel-wise concatenation within the UNet architecture, as introduced in [2]. However,**dedicated DiT-based models for image editing remain unexplored**.
>
> In our empirical analysis, we encountered significant challenges when adapting SD3—a model employing a DiT architecture with a flow-matching objective—to the image editing domain. Specifically, **the channel-wise concatenation approach often compromised SD3’s image generation capabilities, unlike SDXL.** While SD3 benefits from a superior text encoder that enhances its ability to follow instructions and achieve higher semantic consistency (SC) scores, its perceptual quality (PQ) scores were notably lower than those of SDXL.
>
> To address these challenges, we propose a novel EditNet structure that integrates seamlessly with the DiT architecture without modifying its core weights. EditNet introduces a control branch to effectively incorporate image condition signals, **significantly enhancing the quality of the generated content.** This improvement is reflected in the superior PQ scores achieved by EditNet, demonstrating the effectiveness of the EditNet structure in the image editing domain:
>
>
> |       |Gemini(SC↑)|Gemini(PQ↑)|Gemini(Overall)| GPT4o(SC↑)| GPT4o(PQ↑)| GPT4o(Overall)|
> |--------------|---------|---------|------|---------|---------|------|
> | OmniEdit(SDXL)       | 8.23       | 5.88    | 5.97        | 6.91    |   5.12  | 4.82 |
> | OmniEdit(SD3)        | 8.37       | 5.71        | 5.96        | 7.06   | 4.98        | 4.82         |
> | OmniEdit(SD3-EditNet)| **8.38**   | **6.66**    | **6.98**    | **7.06**    | **5.82**        | **5.78**  |
>
> [1] Brooks, T., Holynski, A., & Efros, A. A. (2023). Instructpix2pix: Learning to follow image editing instructions.
>
> [2] Ku, M., Jiang, D., Wei, C., Yue, X., & Chen, W. (2023). Viescore: Towards explainable metrics for conditional image synthesis evaluation.

---

### Meta-Review · Area_Chair_R3yM · 2024-12-16

**Metareview:**

This paper presents an instruction-based image editing method for seven different tasks, trained on the basis of pre-trained task-specific models, as well as LLM scoring. A novel architecture is presented and trained using images with different image resolutions. The authors release a new dataset to the community along with this work.  Initially there were some concerns regarding the evaluations and ablations of the method, and similarilty to recent related work. The paper writing, results, dataset, and proposed approach are appreciated by the reviewers.

**Additional Comments On Reviewer Discussion:**

The authors provided an extensive rebuttal and updates to the manuscript to address most concerns raised by the reviewers. Taking this into account four of five reviewers recommended acceptance. The one negative recommendation comes from a reviewer that did not respond to the author rebuttal. The AC thinks the author rebuttal addresses their main concern (about novelty wrt to a recent cvpr'24 paper) sufficiently.

---

### Decision · Program_Chairs · 2025-01-22

Accept (Poster)